# Exosome-mediated stable epigenetic repression of HIV-1

Surya Shrivastava [1], Roslyn M. Ray [1], Leo Holguin [1], Lilliana Echavarria [1], Nicole Grepo [1], Tristan A. Scott [1], John Burnett [1,2] & Kevin V. Morris [1,2,3 ✉]

Human Immunodeficiency Virus (HIV-1) produces a persistent latent infection. Control of HIV-1 using combination antiretroviral therapy (cART) comes at the cost of life-shortening side effects and development of drug-resistant HIV-1. An ideal and safer therapy should be deliverable in vivo and target the stable epigenetic repression of the virus, inducing a stable "block and lock" of virus expression. Towards this goal, we developed an HIV-1 promoter-targeting Zinc Finger Protein (ZFP-362) fused to active domains of DNA methyltransferase 3 A to induce long-term stable epigenetic repression of HIV-1. Cells were engineered to produce exosomes packaged with RNAs encoding this HIV-1 repressor protein. We find here that the repressor loaded anti-HIV-1 exosomes suppress virus expression and that this suppression is mechanistically driven by DNA methylation of HIV-1 in humanized NSG mouse models. The observations presented here pave the way for an exosome-mediated systemic delivery platform of therapeutic cargo to epigenetically repress HIV-1 infection.

---

[1] Center for Gene Therapy, City of Hope-Beckman Research Institute, Duarte, CA, USA. [2] Hematological Malignancy and Stem Cell Transplantation Institute at the City of Hope, Duarte, CA, USA. [3] Menzies Health Institute Queensland, School of Medical Science Griffith University, Gold Coast Campus, Brisbane, Australia. ✉email: kmorris@coh.org

With the failure of the recent HIV-1 vaccine trial[1] and ART inefficacies, such as treatment failure following drug resistance in up to 22%, poor compliance in up to 46%, adverse effects in up to 83% of people living with HIV-1, and poor central nervous system penetration[2–4], there is a need to re-evaluate the strategies to combat HIV-1 which causes AIDS[5]. Daedal features of the HIV-1 genome and infection cycle enable it to evade the immune system and drug therapies by entering into a latent state[6]. Even though the latent virus is refractory to cART, it could still be a target for therapeutic intervention. Clinical trials and other ex vivo studies on latency-reversing agents (LRAs) such as Histone Deacetylase inhibitors have led to non-specific immune suppressive effects on cytotoxic CD8 + T cells. In the absence of CD8 + T cell-mediated killing of infected CD4 + T cells, the eradication of viral reservoirs has remained subpar[7]. In several clinical trials "shock/kick and kill" strategies have resulted in enhanced viremia but there was no significant elimination of HIV-1 reservoirs[8,9]. Therapeutics has also been developed to lock the virus into a permanent state of latency and block it from re-emergence. Blocking of HIV-1 Tat-dependent transcription by a specific inhibitor didehydro-cortistatin A (dCA) delayed viral rebound but led to the emergence of dCA-resistant virus[10]. Several other drugs which inhibit host cellular partners of HIV-1 transcription like AUY922, 17 AAG, ruxolitinib, tofacitinib, and triptolide have all been shown to block viral transcription and promote latency[11]. However, these drugs also exhibit toxic side effects due to their interaction with cellular proteins[11].

In order to minimize the toxicity, we reasoned that proteins from cellular epigenetic arsenal like DNA Methyl Transferase 3 A (DNMT3A) could be adapted to down-regulate HIV-1 transcription. Hypermethylation of the viral 5′ LTR has been reported to cause latency in cell models of HIV-1 infection as well as within the latent reservoir of some long-term ART-treated individuals[12,13]. Therefore, targeted DNA methylation of the 5′ LTR presents a promising avenue for a "block and lock" strategy. To enrich DNA methylation specifically, the DNA methyl transferase was fused to Zinc Finger Protein ZFP-362b which binds to site 362 on HIV-1 LTR. Previous CRISPR-based screening identified this unique site 362 located between two NFκB binding sites on HIV-1 LTR[14] and highlighted its importance in achieving specific gene regulation without causing non-specific global gene expression. A recent publication from our lab describes ZFP362 that can bind to responsive LTR 362 site and potently activate all clades of HIV-1 with a characterized off-target profile[15]. However, an ideal "block and lock" drug need not only be HIV-1 specific but also have short-lived pharmacokinetics, to prevent off-target disruption at the cellular level. They also need to penetrate systemically to reach multi-tissue HIV-1 reservoirs and be able to induce long-term virus inactivation[16]. Exosomes offer such an approach whereby specific therapeutic moieties can be packaged into discreet, pervasive, nanosized particles which are well tolerated by the immune system[17,18]. Exosomes with natural cargo have been used in diagnostics or as therapeutics[19]. However, packaging exosomes with a choice of cargo has proven challenging owing to their miscellaneous endogenous pathways of sorting cargoes[20]. Kojima et al. recently published an engineered pathway to package selected RNA cargo into exosomes[21]. We took advantage of this exosome system to deliver mRNA for an epigenetic inhibitor of HIV-1 and report herein that these exosomes are capable of inducing an epigenetic "block and lock" of HIV-1 expression in vivo. The observations presented here provide a proof of principle demonstration for the utility of exosome engineering as a means to introduce ephemeral epigenetics-based therapeutics to induce stable targeted modulation of gene expression.

## Results

**The LTR targeted ZFP 362b fused with DNMT3A potently represses HIV-1 expression.** To achieve stable transcriptional repression of HIV-1, a vector expressing a recombinant fusion protein, ZFP 362b-DNMT3A (ZD3A), was developed which can localize to the nucleus with the help of an embedded nuclear localization signal (NLS) (Fig. 1a). This ZFP is capable of binding to the NF-kB doublet binding site within the LTR promoter of HIV-1 and was fused to a DNA methyl transferase 3 domain (3 A) to generate (ZD3A). Transfection of ZD3A into chronically HIV-1-infected Jurkat cells (CHI-Ju) led to significantly reduced HIV-1 transcript expression when contrasted with the control ZFP without DNMT3A (Fig. 1b), an observation which was more pronounced when cells were activated with TNFα prior to transfection (Fig. 1c). Furthermore, a potent reduction of viral transcription was observed 10 days post-transfection showing stable repression by ZD3A (Fig. 1d). From the culture supernatant of transfected CHI-Ju cells, viral RNA copy number[22] and viral p24 protein were estimated (Supplementary Fig. 1a, b) which corroborated well with Gag RNA transcriptional levels. ZD3A-mediated repression appeared to take time to build up and becomes highly significant 10 days later as seen in a time-course experiment (Supplementary Fig. 1c). Next, we constructed several fusion constructs of ZFP362 with various repressor domains; KRAB[23], the active domains of DNMT3A such as ADD, PWWP, Methyltransferase (Mtase) core domain[24] or a catalytic methyl transferase domain of DNMT3A (cdDNMT3A) previously reported in the literature[25]. This reported cdDNMT3A used a catalytically dead Cas9 (dCas9) CRISPR/Cas system, dCas-DNMT3A[25], and was co-transfected with a previously reported 362 site-directed F2 guide RNA[14] (Fig. 1e). ZFP fused to PWWP, ADD, and Methyltransferase (ZPAMt); and ZFP fused to KRAB and Methyltransferase (ZKMt) domains effectively repressed viral expression compared to ZFP-D3A and the ZFP control (Fig. 1f). Expression of ZPAMt and (ZKMt) was confirmed by western blot (Fig. 1g). In an additional in vitro model of HIV-1 infection, microglial cells transformed to express GFP under the HIV-1 LTR promoter[26] were assessed and over-expression of both the ZPAMt and ZKMt significantly repressed GFP expression, further delineating the suppressive nature of these recombinant proteins (Fig. 1h). These observations delineate that the recombinant fusion protein, ZPAMt, and ZKMt repress viral expression.

**ZFP targeted DNA methylation to the HIV-1 LTR.** To determine the mechanism of action involved in the observed silencing, we transfected cells with the ZPAMt and ZKMt and confirmed a 40 to 60-fold increase in over-expression of ZFP mRNA (Fig. 2a). We further confirmed by chromatin immunoprecipitation (ChIP), the binding of these proteins to the HIV-1 promoter (Fig. 2b), which was not observed at the nonspecific GAPDH promoter region (Fig. 2c), demonstrating targeted binding of the LTR by the ZFP-362. In transfected CHI-Ju cells, ZPAMt and ZKMt were observed to potently repress Gag RNA expression (Fig. 2d) and increase targeted enrichment of CpG methylation at the HIV-1 LTR region relative to controls as detected by methyl DNA Immunoprecipitation (meDIP) assay (Fig. 2e). The observed CpG methylation was specific, as DNA pulldown for CpG methylation did not show amplification at the downstream Tat region of HIV-1 (Fig. 2f). It is noteworthy that the ZFP control also localizes to the 362 sites but does not appear to suppress the virus or enrich CpG methylation to an extent comparable to the fusion constructs (Fig. 2e). These data confirm that the observed suppression is the result of ZPAMt and ZKMt directed DNA methylation to the HIV-1 LTR.

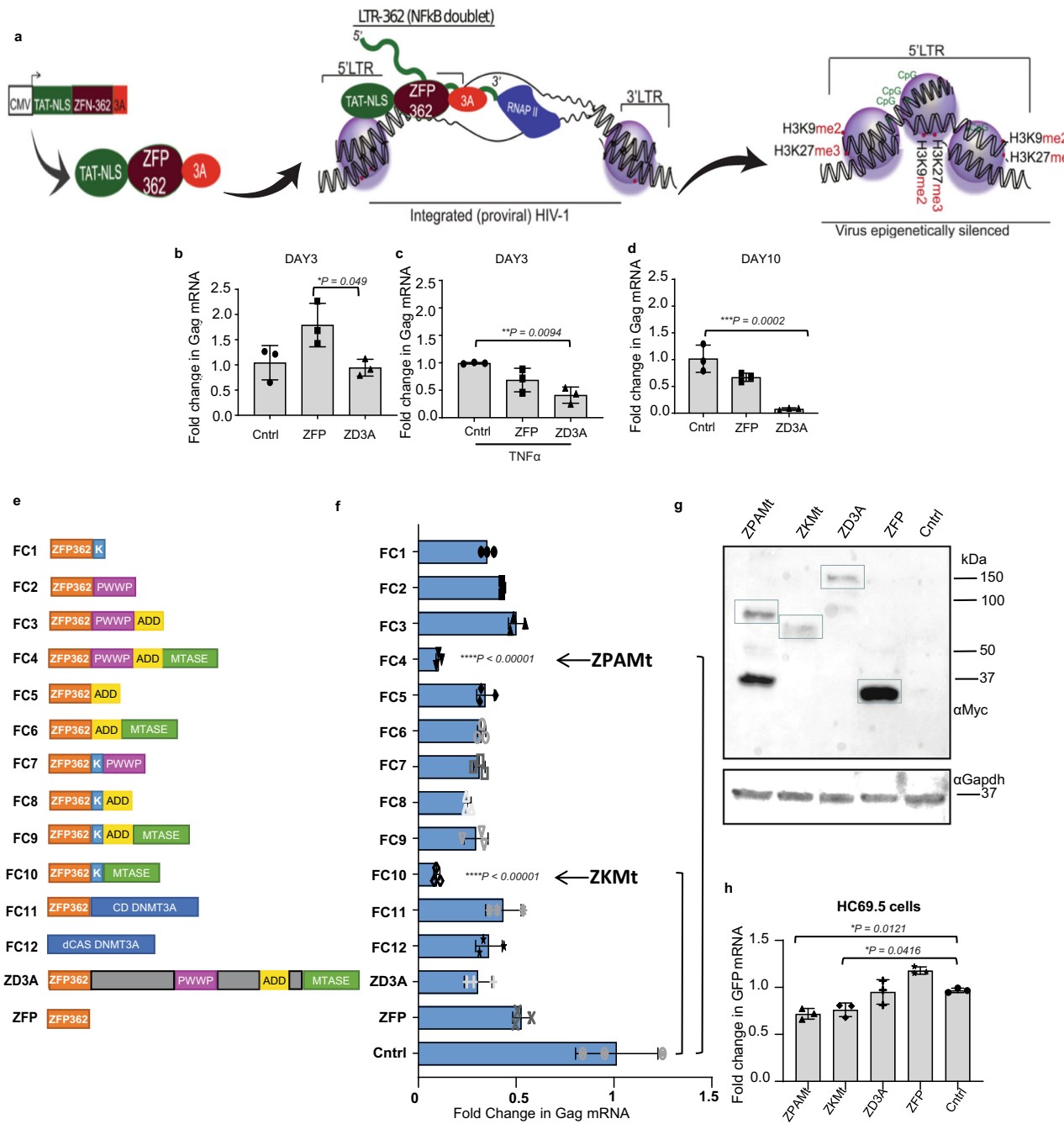

**Fig. 1 Screening of potent repressors of HIV-1 derived from fusion of ZFP362 and combination of repressor domains. a** Schematic representation of the design of the ZD3A constructs leading to ZD3A protein. The NLS containing the ZFP362 domain can bind HIV-1 LTR promoter specifically and DNMT3A can recruit epigenetic silencing complexes. The binding of ZFP362-DNMT3A (ZD3A) can lead to long-term epigenetic transcriptional repression of the LTR represented by methylated CpG DNA and repressive histone methylation (right). **b** The effect of transfection of ZFP and ZD3A constructs in chronically infected Jurkat (CHI-Ju) cells was measured via RT-qPCR for Gag mRNA at 3 days post-treatment. **c** Measurement of Gag mRNA from TNFα activated CHI-Ju cells three days after transfection with ZD3A. **d** The effect of transfection of constructs ZFP and ZD3A in chronically infected Jurkat (CHI-Ju) cells 10 days post-treatment was measured by RT-qPCR for Gag mRNA. **e** A schematic is shown here depicting the 12 different constructs developed and assessed consisting of ZFP362 with one or more repressor domains like KRAB(K), PWWP, ADD, Methyltransferase (Mtase), Catalytic Domain (CD) of DNMT3A fusions, and the full-length DNMT3a (ZD3A) and control ZFP-362. **f** CHI-Ju cells were transfected with those fusion constructs depicted in (**e**) and RT-qPCR for Gag mRNA 10 days post-transfection revealed FC4 and FC10 to be the most potent repressors of LTR activity. FC4 will henceforth be referred to as ZPAMt after constituent ZFP362, PWWP, ADD, and Mtase domains, and FC10 will be referred to as KMt after constituent ZFP362, KRAB, and Mtase domains. **g** Western blot for Myc tag confirmed the expression of ZPAMt, ZKMt, ZD3A, and ZFP. Representative blot of *n* = 3 biological replicates (**h**) LTR driven GFP mRNA expression level measured after overexpression of ZPAMt and ZKMt in microglial cell line HC69.5 Data are represented as mean ± standard deviation (SD) in **b–d**, **f**, and **h**. *p*-values by one-way ANOVA followed by Tukey's multiple comparisons post hoc test. For **b–d**, **f**, and **h** the error bars indicate the standard deviation of triplicate treated samples and experiments were repeated twice. ****p ≤ 0.0001. Source data are provided as a source data file.

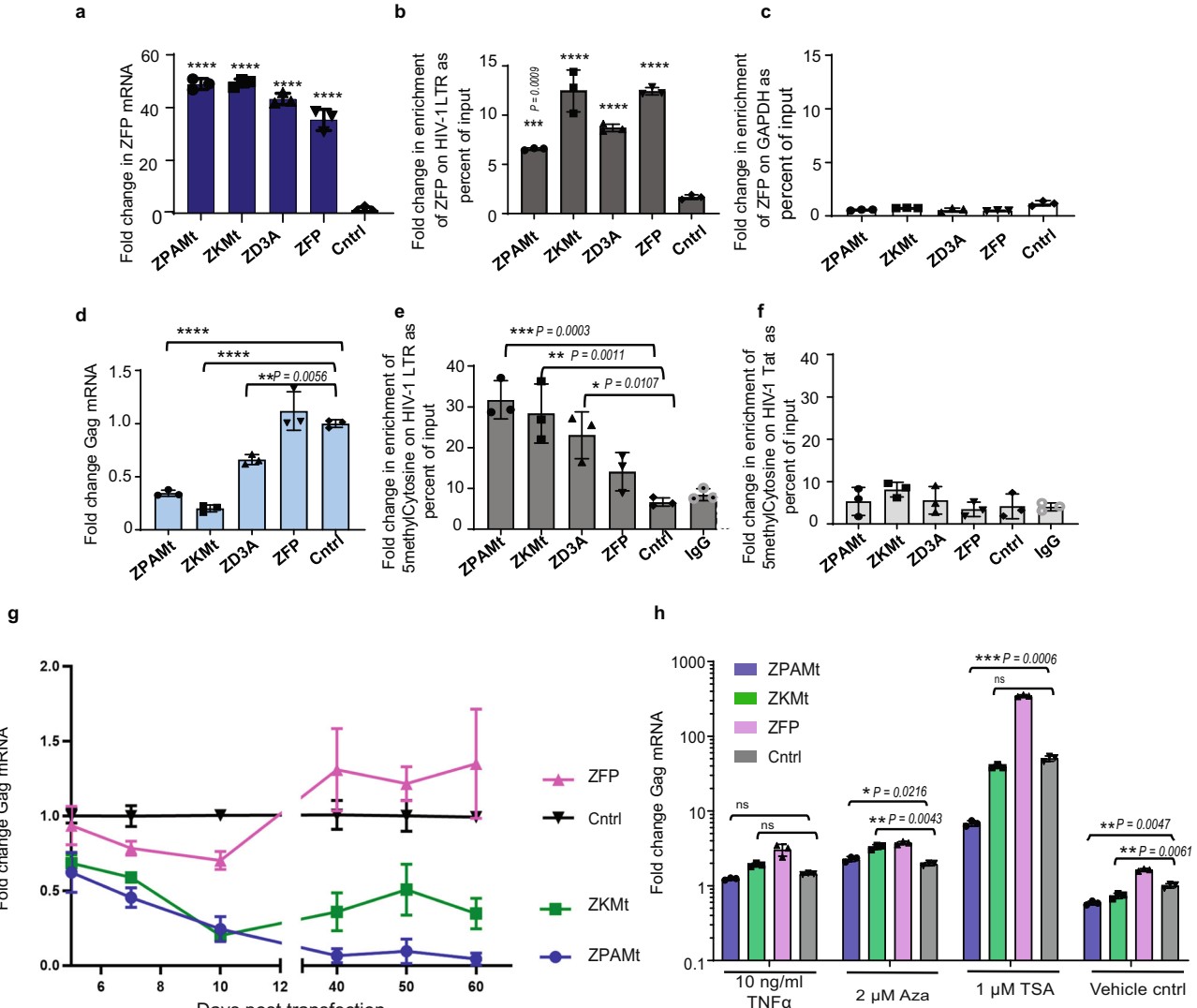

**Fig. 2 ZPAMt and ZKMt occupy and methylate HIV-1 LTR promoter and repress virus expression long-term manner that is refractory to activation.**
**a** RT-qPCR for ZFP mRNA, two days post-transfection in CHI-Ju cells reveals overexpression of mRNA from ZFP362 fusion constructs or pcDNA control (cntrl). **b**, **c** ChIP with Myc tag beads reveals (**b**) occupancy of ZFP362 fusion proteins and ZFP362 alone on HIV-1 LTR and (**c**) little to no binding to the non-specific GAPDH region. **d** RT-qPCR for Gag mRNA, ten days post-transfection of constructs in CHI-Ju cells reveals potent HIV-1 transcription repression from ZFP362 fusion constructs compared to ZFP362 backbone. **e**, **f** MeDIP assay from cells transfected with fusion constructs show (**e**) enhancement of CpG methylation on LTR (**f**) no significant changes in CpG methylation on Tat region of NL4-3 genome. For **a**–**f** the data represents means ± SD of triplicate treated samples and experiments were repeated twice. *p*-values were determined by one-way ANOVA followed by Tukey's multiple comparisons post hoc test. ****$p \leq 0.0001$. **g** Treated cultures were assessed by RT-qPCR for GagGag mRNA for two months post-transfection of ZPAMt and ZKMt. **h** RT-qPCR for Gag mRNA shows a level of reactivation of HIV-1 transcripts from transfected CHI-Ju cells after treatment with various latency reactivation reagents and 0.1% DMSO was used as vehicle control. Each experimental sample was compared to pcDNA transfected control. For **g**, **h** data represent mean ± SD of triplicate treated samples. For **h** *p*-values were determined by two-way ANOVA followed by Tukey's multiple comparisons post hoc test. ****$p \leq 0.0001$ Source data are provided in the source data file.

As CpG methylation is a long-term epigenetic mechanism of gene silencing[27], we determined the duration of silencing of both ZPAMt and ZKMt and found that both constructs exerted repression in CHI-Ju cells for 60 days post-transfection (Fig. 2g). To determine the potency of "block and lock" of the LTR induced by the targeted CpG methylation, we screened various latency reactivation agents Azacytidine (Aza), Trichostatin A (TSA), and TNFα[28] for their ability to reactivate the stably repressed virus after treatment with ZPAMt and ZKMt. In TNFα and Aza-treated samples, Gag mRNA expression from ZPAMt and ZKMt transfected samples were either comparable to or higher than, the control transfected samples (Fig. 2h). Notably, treatment with TSA in those cells harbouring the ZPAMt-repressed virus,

demonstrated a lag in reactivation compared to the control. This observation signified that ZPAMt-mediated repression is refractory to activation in the presence of HDAC inhibitors, as would be expected based on the known mechanisms of DNA methylation. Supporting this notion, was the observation that in the presence of TSA, the levels of Gag mRNA expression from ZKMt transfected cells remained insignificantly different from the control (Fig. 2h). Collectively, these studies suggest that ZPAMt stably represses HIV-1 in a long-term and specific manner by targeting CpG methylation of the 5' LTR and that this repression is relatively resistant to reactivation. As a result, we selected ZPAMt and explored the possibility to make it deliverable to virus-infected cells in vivo via exosomes.

**Exosome-packaged ZPAMt functionally represses virus expression in CHI-Ju cells and human peripheral blood mononuclear cells**. The EXOtic (EXOsomal transfer into cells) system was adopted[21] to package ZFP or ZPAMt into cellular exosomes. In this EXOtic system, the ubiquitous exosome marker protein CD63 is fused to an archaebacterial derived L7Ae peptide, which allows for the recruitment and encapsulation of those mRNAs containing C/D$_{box}$ to budding exosomes. The DNA sequence for C/D$_{box}$ was cloned at the 3'end of ZFP and ZPAMt to express mRNA containing the C/D$_{box}$ RNA domain. The production of exosomes was boosted using a tri-ORF plasmid system containing STEAP3-IRES-SDC4-IRES-nadB. Additionally, mutant protein Connexin43 S368A was over-expressed to increase the release of the ZFP payloads into target cells[29] (Fig. 3a). To address the hypothesis that therapeutic ZFPs can be delivered to virus-infected cells via exosomes, we characterized the isolated exosomes by Nanosight tracking analysis (NTA) and transmission electron microscopy (TEM). Exosomes packaged with ZFP or ZPAMt were found to be comparable in size, with an average of ~100 nm diameter, and morphology to control exosomes loaded with nano luciferase (nLuc) (Fig. 3b). Exosome markers CD63, Alix and TSG101 were detected from lysates of exosomes derived from HEK293T cells, transfected with the EXOtic plasmid device, which confirmed their purity and identity as exosomes (Fig. 3c). Using linearized ZFP DNA construct as a standard, copies of mRNA packaged per $10^8$ exosomes were calculated by RT-qPCR. Nearly 20,000 RNA copies from ZFP loaded $10^8$ exosomes and 7000 ZFP mRNA copies from ZPAMt loaded $10^8$ exosomes were detected. Our results reveal efficient incorporation of RNA containing ZFP within exosomes (Fig. 3d). To determine if these ZFP mRNA containing exosomes could disseminate protein to target cells, immuno-blot from HEK293T cells exposed to the candidate exosomes confirmed the presence of myc-tagged ZFP or ZPAMt proteins in recipient cells (Fig. 3e). Next, EXOtic device-derived exosomes were packaged with either reporter nLuc, ZFP, or ZPAMt mRNA and added directly to CHI-Ju cells to assess the effect of exosome-mediated transfer of the ZPAMt repressor in a virus-infected cell system. Significant repression in Gag mRNA expression was observed when a ratio of 10,000 exosomes per cell was used in the assay (Fig. 3f). Equal numbers of CHI-Ju cells were seeded, treated with exosomes in a ratio of $10^4$ exosomes per cell every alternate day for three doses and then viral copy numbers from 100 μl of supernatant from equal numbers of cells were measured via RT-qPCR (Supplementary Fig. 2a). Viral copy number corroborated well with cellular Gag RNA measurement and demonstrated significantly lower virus expression due to ZPAMt exosome treatment compared to control or ZFP. Latent cell lines U1 and ACH2 were activated with 10 ng/ml TNFα treatment for 24 h and washed before treatment with exosomes. Similar treatment and measurement of HIV-1 copy number revealed significantly lower HIV-1 production from these two cells lines following treatment with ZPAMt-packed exosomes (Supplementary Fig. 2b, c). To determine if these exosomes are functional in the setting of primary cells infected with HIV-1, peripheral blood mononuclear cells (PBMCs) were infected with HIV-1 and then were exposed to exosomes. Treatment with ZPAMt exosomes in this setting exhibited significant declines in HIV-1 transcription in 4 out of 5 donors PBMCs tested (Fig. 3g). Collectively, these data demonstrate that ZPAMt can be packaged into exosomes using the EXOtic device and disseminated through exosome-mediated delivery to target cells whereby the ZPAMt can functionally suppress HIV-1 transcription.

**A stable Lentivirus vector-transduced therapeutic exosome producer cell system**. To reduce the need of frequent transfections we developed a stable therapeutic exosome secreting producer cell system. Lentiviral vectors (LV) generated from producer lentiviral plasmids with EXOtic device components enabled cells to stably produce exosomes with desired cargo as shown in Supplementary Fig. 3a. LV derived from plasmid for producer pLV-EXOtic-ZPAMt was used to transduce cells so that they can express CD63-L7Ae and Connexin43 S368A separated by P2A site protease cleavage site. These cells also overexpressed ZPAMt-C/D$_{box}$ mRNA under the EF1α promoter. Similarly, LV-EXOtic-ZFP and LV-EXOtic-nLuc-transduced cells were made to stably express EXOtic components and exosomes packaged with ZFP or nLuc. Booster plasmid was also adopted to be packaged as a separate LV. Both HEK293T cells and hTERT immortalized MSC were transduced with either of the three exosome producer LV along with LV-Booster and the transduced MSC were cultured. To study the effect of transfection of EXOtic plasmid device versus transduction of EXOtic LV, HEK293T cells were transfected with EXOtic component plasmids or transduced with EXOtic producer LV and booster LV. Exosomes were harvested from HEK293T cells either after transfection with EXOtic device plasmids or after stable EXOtic LV transduction and added on CHI-Ju cells. Owing to quick uptake and short half-life of exosomes[30], a repeat dosing strategy for enhancing the effect of ZPAMt was also tested. We found that three doses of ZPAMt loaded exosomes were more effective compared to a single dose in repressing HIV-1 transcription (Supplementary Fig. 3b). Additionally, both transiently transfected and stable producer cell-derived exosomes significantly repressed HIV-1 Gag mRNA expression in CHI-Ju cells (Supplementary Fig. 3b). The producer lentivirus with either nLuc (Control), ZFP, or ZPAMt as a payload and booster lentivirus were used to transform hTERT immortalized mesenchymal stem cells[31]. The transduced producer MSCs were seeded in 12-well plates, on top of which CHI-Ju cells were added in a transwell with 0.4 mm size pores to enable the free exchange of exosomes (Supplementary Fig. 3c). When ZPAMt exosomes producing MSCs were co-cultured with CHI-Ju, repression of HIV-1 Gag mRNA was observed (Supplementary Fig. 3d). Though the parent hTERT immortalized MSC stably express firefly luciferase (fLuc), the transduced producer MSC-derived exosomes were shown to transfer specifically packaged nLuc signals and not passively packaged fLuc signals to the recipient cells (Supplementary Fig. 4). Interestingly, we observed that transiently transfected HEK293T were more robust in producing therapeutic exosomes which were also notably more potent than stably transformed producer cell-derived exosomes in both HEK293T (Supplementary Fig. 3b) and stably transduced MSCs (data not shown). As such transiently transfected HEK293T derived exosomes were thus chosen for in vivo delivery of reporter control or therapeutic cargo.

**Exosome-packaged ZPAMt represses virus expression in vivo**. To determine the in vivo biodistribution of the reporter exosomes, $20 \times 10^9$ exosomes were injected via two different routes of injection in NSG mice, intraperitoneal (IP) or retro-orbital (RO). After 4 h, mice were euthanized and luciferase activity from lysates of the bone marrow, brain, spleen, and liver were quantified. Luciferase activity was higher in bone marrow, brain, and liver and lower in the spleen when exosomes were delivered via RO compared to IP (Supplementary Fig. 5a). We also RO-injected nLuc packaged exosomes in two different doses of 20 and $140 \times 10^9$ exosomes per mouse, and four hours later nLuc activity was detected from brain, bone marrow, liver, and spleen lysates. The increase in nLuc activity was concordant with the dose (Supplementary Fig. 5b). These observations led us to select the dosage of $100 \times 10^9$ exosomes and RO delivery route to study the

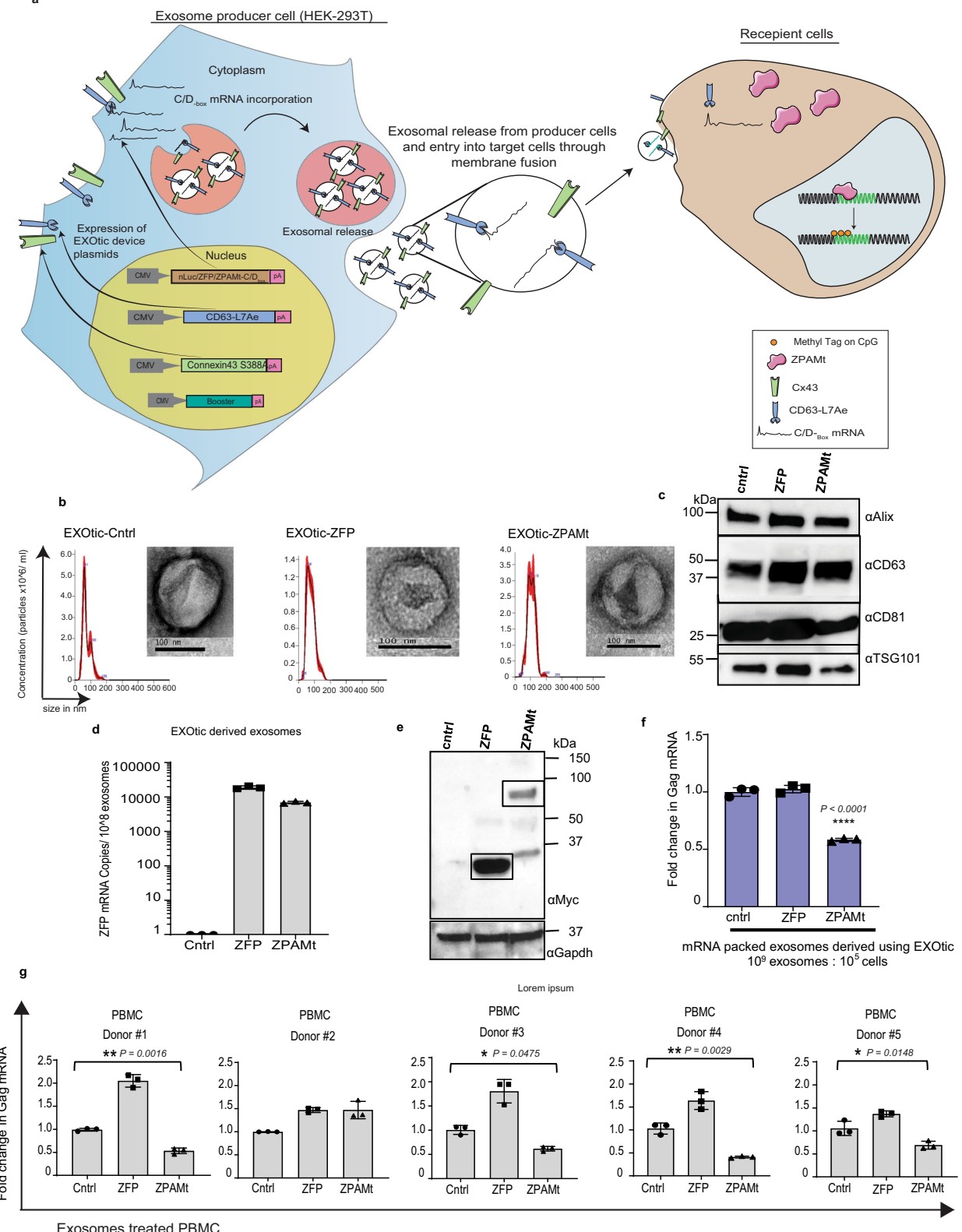

efficacy of ZPAMt to control HIV-1 in both the hu-PBMC-NSG and hu-CD34 + NSG mouse models.

A preliminary study was done to evaluate the efficacy of oral cART administration on hu-PBMC NSG mice. After confirmation of infection at week 2 post-HIV-1-1 injection, oral cART was administered for four weeks. Efficient suppression of HIV-1-1 (Supplementary Fig. 6a) and rescue of human CD4 + cells

(Supplementary Fig. 6b) were observed within 4-weeks of oral cART administration. Due to the observed loss of CD4 + cells 'no cART' mice demonstrated a decline in viremia while in the oral cART treated mice rapid viremia was observed after 9 weeks due to resurgence of HIV-1 (Supplementary Fig. 6a). These data demonstrate that a rapid rise in viremia and acute loss of CD4 + cells is observed after oral cART was stopped. To prevent

**Fig. 3 Exosome mediated delivery of ZPAMt mRNA to virus infected cells. a** Schematic for mechanism ZPAMt mRNA transfer from producer cells to recipient cells. HEK293T producer cells are transfected with the following 4 plasmids: "Booster" to increase exosome production, cytosolic delivery helper "Connexin43 S368A", packaging plasmid CD63-L7Ae, and cargo mRNA-C/D$_{box}$ plasmid. The cargo was intended to be either nLuc reporter, ZFP (non-fusion control), or ZPAMt. Exosomes derived from these producer cells with the ZPAMt-C/D$_{box}$ mRNA can transfer ZPAMt-C/D$_{box}$ mRNA in recipient cells and produce the ZPAMt fusion protein. This ZPAMt protein leads to transcriptional epigenetic repression of HIV-1 within the infected cell. **b** Concentration and size distribution of exosomes secreted from producer cells transfected with EXOtic system plasmids with either control nLuc-C/D$_{box}$, ZFP-C/D$_{box}$, or ZPAMt-C/D$_{box}$ cells engineered with the exosome production booster as measured by Nanoparticle tracking analysis (NTA). TEM image insets show the morphology and size of EXOtic derived exosomes. Scale bars 100 nm. NTA graphs and TEM images representative of $n = 3$ biological replicates. **c** Western blot analysis from exosome lysates shown the expression of common exosome protein markers like Alix, CD63, CD81, and Tsg10. Representative blot of $n = 2$ biological replicates. **d** Copy number calculation by RT-qPCR for ZFP region reveals packaging of ZFP/ZPAMt mRNA content in exosomes compared to nLuc mRNA packaged exosomes used as a control. **e** Western blot analysis using an anti-Myc antibody of lysates from ZFP and ZPAMt exosome-treated cells. GAPDH was detected as a loading control. Representative blot of $n = 3$ biological replicates **f** RT-qPCR for Gag mRNA was detected in CHI-Ju cells treated with exosomes containing either nLuc (control), ZFP, or ZPAMt. **g** Gag expression after three doses of HEK-293T derived exosomes on five separate healthy donor-derived, HIV-1-infected PBMCs. PBMCs were treated with exosomes with nLuc loaded RNA to serve as a control. The error bars indicate the standard deviation of triplicate-treated samples. For **f, g** $P$ values were determined by one-way ANOVA followed by Tukey's multiple comparisons post hoc test. ****$p \leq 0.0001$. Source data are provided in a source data file.

acute HIV-1-1 causing the systemic loss of human CD4 + cells in HIV-1-1 infected hu-PBMC NSG mice when evaluating the efficacy of ZPAMt, short-term oral ART administration was deemed indispensable.

To determine the antiviral efficacy of ZPAMt, humanized NSG mice were engrafted with HIV-1-infected PBMCs (hu-PBMC-NSG) followed by the administration of oral cART for two weeks and subjected to six exosome doses injected weekly (Fig. 4a). Prior to treatment, at the initial 2-week time point, the viremia in HIV-1-infected hu-PBMC-NSG mice was not found to significantly differ and as such they were divided into three groups that received either nLuc, ZFP, or ZPAMt-packaged exosomes. Mice receiving the ZPAMt exosomes significantly repressed HIV-1 expression when contrasted with the ZFP or nLuc exosome-treated mice (Fig. 4b). Following the guidelines to combine preliminary and new animal data[32], median HIV-1 viremia levels in each group i.e. oral cART-treated, 2 weeks (2 W) cART + nLuc exosome-treated, 2 W cART + ZFP exosome-treated, 2 W cART + ZPAMt exosome-treated, and 'no cART' mice were plotted in one graph (Fig. 4c). Only ZPAMt exosome-treated mice were found to demonstrate significant stable repression out to the final time point of week 10. At the experimental endpoint (week 10) mice were euthanized, and viremia levels were measured in the bone marrow and spleen. A notable trend of lower HIV-1 Gag transcripts was observed in both tissues in ZPAMt exosome-treated mice (Supplementary Fig. 7a, b). To determine the mechanism underlying the observed repression of HIV-1 in vivo, CpG methylation was assessed at the 5' LTR in human CD45 + cells isolated from the bone marrow of HIV-1-infected hu-PBMC-NSG. Significant methylation was observed in the ZPAMt exosome administered mice compared to the control groups (Fig. 4d). Notably, the ZPAMt exosome treatment had no off-target effect on DNA methylation of the human GAPDH promoter (Supplementary Fig. 8a), or the human histone 2B type1A (H2B1A) region (Supplementary Fig. 8b). Methylation enrichment at H2B1A served as a positive control for the meDIP assay suggesting that ZPAMt targeting of the 5' LTR is specific. Alanine transaminase (ALT) and aspartate aminotransferase (AST) assays did not demonstrate any significant differences between untreated NSG mice and designer exosome-treated mice (Fig. 4e), revealing that neither HEK293T exosomes nor their respective therapeutic cargo exerted any long-term toxic effects on liver function.

While these studies in hu-PBMC-NSG mice demonstrate that ZPAMt, when delivered via exosomes, can repress HIV-1 expression and impart LTR-specific methylation, we were unable to assess the effects of this treatment on brain HIV-1 reservoirs in

this model. Exosomes have been shown to cross the BBB[33] and we also observed in vivo delivery of therapeutic cargo to the brain (Supplementary Fig. 9). Hence, we wanted to determine if exosomes-packaged ZPAMt could affect the HIV-1 reservoir in the brain, a region that is often inaccessible to cART. As opposed to acute infection progression in the hu-PBMC-NSG model (Fig. 4b), the HIV-1-infected hu-CD34 + NSG model permits chronic infection progression and harbors active HIV-1-infected cells in the brain[34]. After engraftment and establishment of HIV-1 infection in hu-CD34 + NSG mice, two weeks of oral ART therapy was followed by six doses of exosomes (Fig. 5a). As the efficacy of oral cART was already confirmed with the hu-PBMC NSG mouse model (Supplementary Fig. 6), cART-treated or 'no cART' hu-CD34 + NSG mice were not included in these sets of experiments. There was no significant difference in viral load observed in all three groups of mice at two weeks post-infection with HIV-1 (Fig. 5b). Retro-orbital bleeding from mice was done every two weeks to detect viremia. Similar to observations with the hu-PBMC-NSG model, a significant reduction in viremia was observed after administration of six doses of ZPAMt-packed exosomes compared to nLuc or ZFP-exosome-treated mice (Fig. 5b).

After 10 weeks of monitoring viremia, paraffin-embedded brain sections from the NSG-huCD34 + HIV-1 mice were processed for immunohistochemistry to confirm the presence of HIV-1. Using the mouse brain atlas[35], we determined the regions predominantly found in level I, II, and III sections. While p24 + cells were distributed throughout the tissues, we observed that within the level I coronal sections, p24 + cells were concentrated in the cortex and the pyramidal layer in the hippocampal formation region (Supplementary Figs. 10a–b, 11a). In level II coronal sections, p24 + cells were observed within the granular and pyramidal layer of the hippocampal formation region (Supplementary Fig. 10c, d). In level III coronal sections, we observed p24 + cells within the granular layer as well as the Purkinje cell layer in the cerebellum as neuronal cells can also uptake gp120[36,37] (Supplementary Fig. 10e, f). HIV-1 infection in our mouse model progressed for 10 weeks, as such, p24 + staining in this region of the brain may be indicative of late-stage HIV-1 infection, similar to that observed by Wächter et al., (2016) in macaques with SIV infection[38]. We observed p24 + cells in the meninges in addition to the cortex and cerebellum as observed by others[34,39]. RNA was extracted from fixed brain tissue and RT-qPCR for the Pol region using a Taqman probe. We observed that ZPAMt exosomes induced a decline in viral RNA copies in the brain of ZPAMt exosome-treated mice compared to ZFP and nLuc controls (Fig. 5c). Suppression of HIV-1 expression was

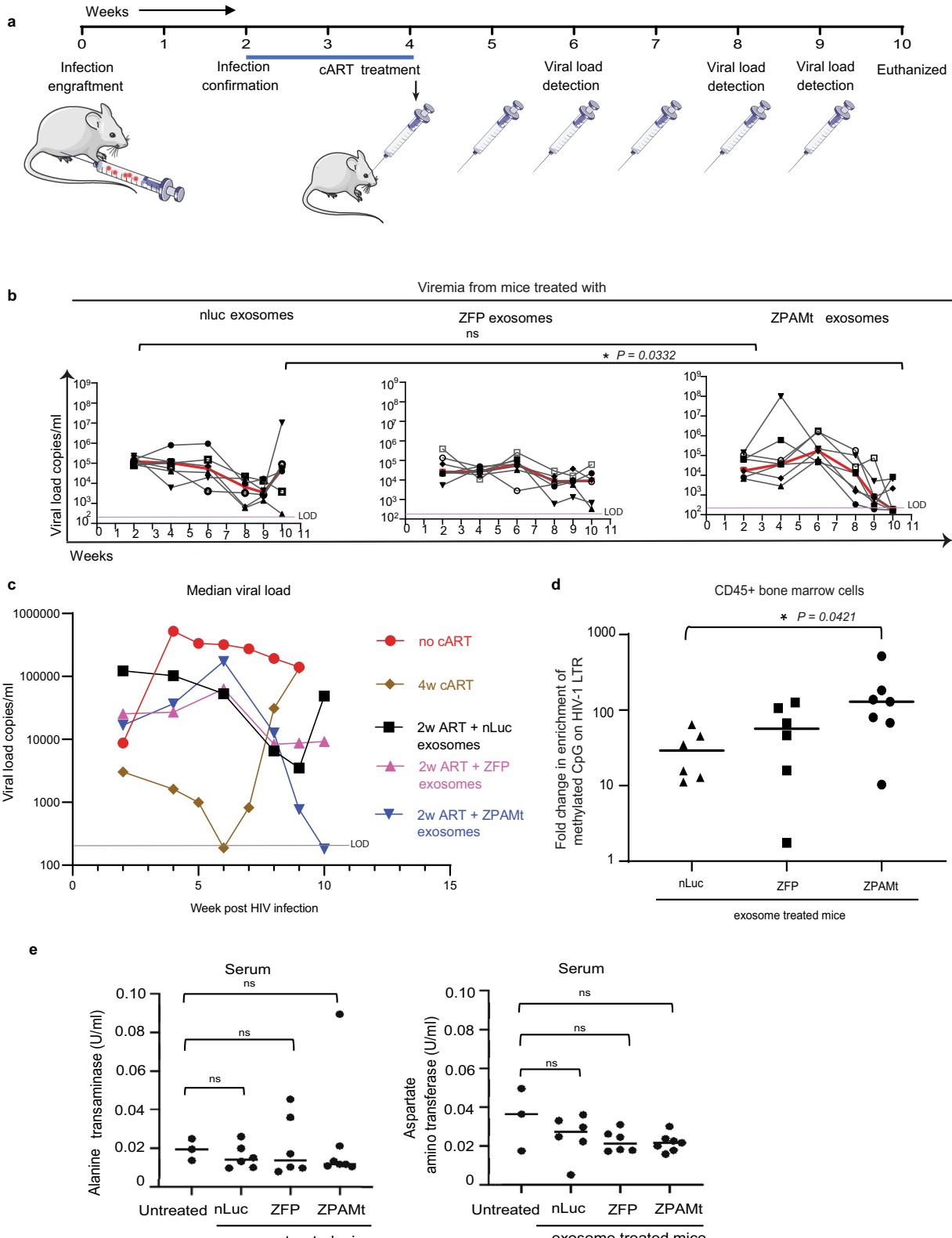

found to be significant in ZPAMt exosome-treated mice compared to controls in the bone marrow and spleen tissue lysates of CD34 + mice (Fig. 5d, e). To assert that significant repression in viremia levels were not due to the dwindling population of CD4 + cells, CD4 + cells were acquired from fixed peripheral blood of mice before the start of treatment and after mice were euthanized. Following the gating strategy shown in Supplementary

Fig. 12a and equating with pre-counted beads, the absolute population of CD4 + cells in hu-PBMC mice (Supplementary Fig. 12b) and hu-CD34 + NSG mice (Supplementary Fig. 12c) were plotted. Significant increases in CD4 + cells were observed in all the mice at week 10 compared to week 2. Higher CD4 + cells were found in the exosome ZPAMt-treated hu-CD34 + mice compared to controls. As CD4 + cells were maintained in both the

**Fig. 4 In vivo distribution of exosome cargo and ZPAMt exosome mediated suppression of viral load in HIV-1-infected hu-PBMC-NSG mice. a** Schematic illustrating the experimental plan to check the efficacy of ZPAMt exosomes in controlling HIV-1 infection progression in hu-PBMC-NSG mice. **b** Dynamics of viremia levels in hu-PBMC-NSG mice after treatment with either nLuc exosomes (left), ZFP exosomes (center), ZPAMt exosomes (right) over a course of 10 weeks of infection and treatment as measured by RT-qPCR. The trendline in red indicates the median of the viral load. The pink line indicates the limit of detection (LOD) of the PCR assay is 200 RNA copies/ml in 50–80 µl of mice plasma. **c** Median values of HIV-1 load monitored periodically in hu-PBMC NSG mice subjected to different treatments i.e. 4 weeks (w) oral cART, 2w oral cART + nLuc or ZFP or ZPAMt exosomes, and no cART mice. Greyline indicates LOD. **d** MeDIP assay revealed the status of DNA methylation as an effect of treatment of nLuc/ZFP/ZPAMt exosomes on BM-derived cells from HIV-1-infected hu-PBMC-NSG mice. **e** Toxicity assay was done by measuring Alanine transaminase activity (left) and Aspartate aminotransferase assay activity (right) for all the mice and was compared to activity from untreated NSG mice. Ordinary one-way ANOVA was used to calculate significance $P > 0.05$ is nonsignificant (ns). In **b**–**e** $n = 6$ was used for nLuc and ZFP treated mice, $n = 7$ was used for ZPAMt treated mice and in (**c**, **e**) $n = 3$ was used for oral cART treated, no cART, hu-PBMC NSG, and untreated NSG. For **b**, **e** $P$ value was determined by the Kruskal–Wallis test followed by Dunn's post hoc test for multiple comparisons. Source data are provided in the source data file.

---

mouse models until the last time point, we found a significant reduction in viremia when baselined with the CD4 + cell count (Supplementary Fig. 12d, e). Overall, these data demonstrated that ZPAMt exosomes administered systemically can penetrate and cross the BBB to deliver anti-HIV-1 therapeutic proteins and effectively combat HIV-1 infection and progression.

## Discussion

Packaging exosomes with a choice of cargo has much therapeutic potential. Proteins have been loaded into extracellular vesicles by adding WW tags on either GFP, Cre, or Cas9[40,41] or miRNA using an exosome sorting sequence[42]. Chemical or electroporation-based loading of exosomes have also been shown to be effective for siRNA or gapmer packing and delivery[43–45]. However, these methodologies are not useful for the packaging and delivery of the ~3 kb ZPAMt mRNA via exosomes. As such, we adapted the EXOtic plasmid device to shift our dependency from the cell's natural but dynamic pathways of sorting cargo into exosomes towards an engineered and robust way of packaging ZPAMt into exosomes. We were able to package the ZPAMt repressor into exosomes and achieve efficient silencing in the recipient HIV-1-infected cells. Low-efficiency ZPAMt packaging of ~1 RNA copy per 10,000 exosomes was compensated by high efficiency of ZPAMt activity and high exosome to cell ratio of 10,000 to one cell, given in multiple doses. This methodology has recently also been adopted by others to package Wnt5a into exosomes to enhance vascularization ex vivo[46]. We have extended the EXOtic device for use in human disease by compacting the four-plasmid system into a two-part lentivirus vector system in order to reduce the transfection requirement in hard-to-transfect cells like MSCs. Although transient transfection in HEK293T was more robust in terms of exosomes production in vitro, stably transduced MSCs can still be administered for conditions where a steady release of therapeutic cargo-loaded exosomes is required. MSC cells stably producing exosomes were also efficient in the repression of HIV-1 in infected Jurkat cells bolstering the utilization of these cells as a source of therapeutic exosomes as proven in clinical trials[47,48]. We also observed here that repeated dosing of $100 \times 10^9$ exosomes in ~25 gram mice resulted in effective silencing of HIV-1. Exosomes at these dosages and frequency did not appear to cause liver toxicity, from physiological build-up akin to previous observations of quick clearance of exosomes via the kidney[49]. In line with this notion of the short-lived nature of exosomes, we observed a waning in the expression of the ZPAMt cargo between 4 h to 1-week post-delivery. However, there was an observed long-term silencing effect mediated even with this relatively short-lived expression of ZPAMt. Surprisingly, we observed an increase in CD4 + cells in all the three exosome-treated groups, suggesting exosomes by themselves can rescue CD4 + cells. Proliferation, activation, or differentiation of

CD4 + cells by exosome treatment has also been extensively established in the literature[50–52]. While varying in methodology and outcome, all these studies have a common underlying theme that exosomes, irrespective of source, boost CD4 + T-cells. These observations are noteworthy as they suggest that exosomes could prove to be an ideal biomolecule delivery vector for the temporal regulation of particular gene pathways involved in human disease.

While ZPAMt demonstrated efficacy against HIV-1 and can be delivered systemically in vivo, a limitation in this therapeutic approach will be the requirement to design a ZFP specific to each HIV-1 subtype. Due to the specificity of ZFP362, we have not tested its potency on other HIV-1 subtypes, although ZFP362 fused to an activator domain has been found to activate clades A-E of HIV-1[15]. Another limitation, rooted in the observation that HIV-1 LTR DNA hypermethylation is not found to be significantly enriched in latent reservoirs from long-term non-progressors of AIDS[53,54], corroborates the findings reported herein that significant DNA methylation does not lead to a consistent reduction of viral load in the mice models.

In general, there exists a lacuna in the literature regarding how DNA hypermethylation drives long-term latency in some HIV-1 patients and not in others and how it can be harnessed and applied to benefit HIV-1 patients. This discrepancy may be in the region of the LTR which is hypermethylated. Notably, the ZFP362 moiety within ZPAMt is designed to bind a unique and conserved NF-κB doublet which is required for viral transcription[55,56]. Targeted silencing of this NF-κB region therefore may limit the ability of the virus to evolve resistance to this form of therapy and reactivation from latency. Even if reactivation of HIV-1 from latency and ZPAMt mediated "block and lock" were to occur, we speculate that ZPAMt can be re-administered as a repetitive dosing strategy without providing a window for the virus to evolve resistance.

Collectively, the studies are shown here demonstrate that the endogenous cellular machinery can be engineered to package and deliver the recombinant ZPAMt to target the epigenetic silencing of HIV-1 and induce a "block and lock" phenotype in virus-infected cells. The anti-HIV-1 therapeutic exosomes presented in this study, with additional preclinical safety studies have the potential to be adopted for clinical trials along with cART for people living with HIV-1, which may reduce the stringency of drug regimen and enhance their quality of life. Furthermore, we have shown a viable route to achieve specific hypermethylation of an integrated provirus, which can be extended beyond HIV-1/ AIDS therapy. Moreover, the results presented here raise the potential of exosomes to function as next-generation delivery vehicles for epigenetic modulators of specific genes and harken to a future whereby tailored exosome therapeutics could be envisioned to be deployed to control various infectious diseases and disease-relevant genes.

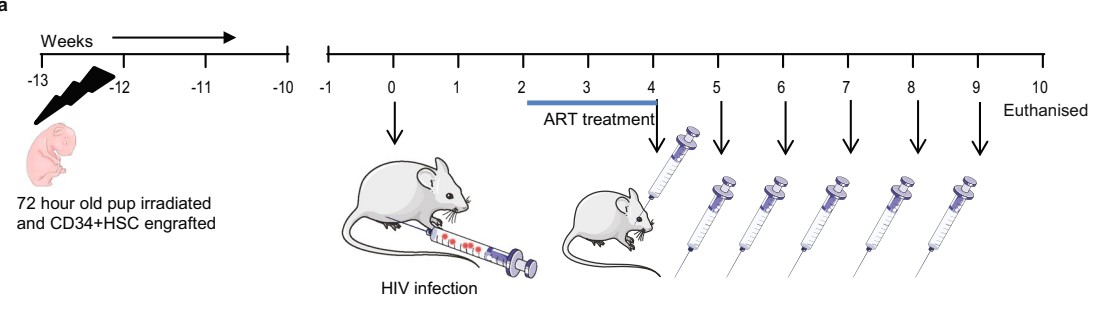

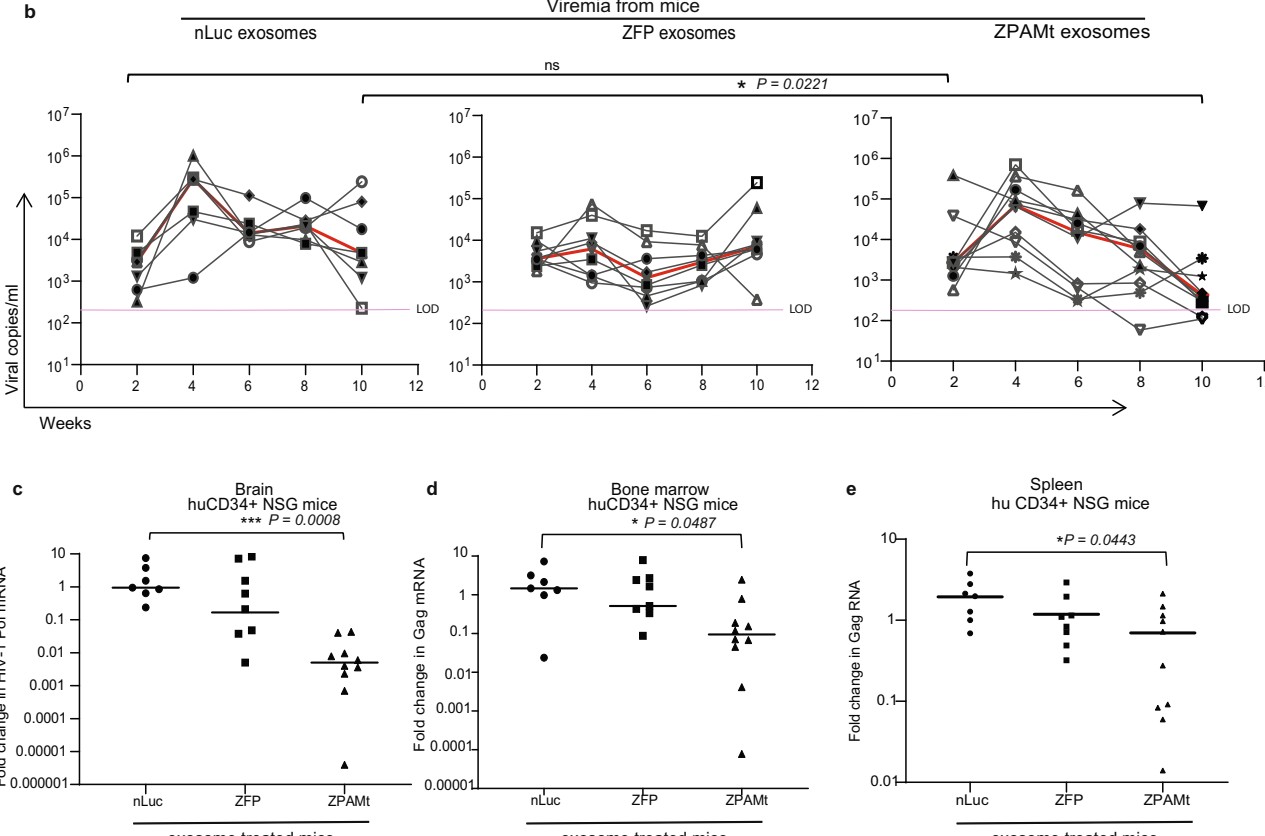

**Fig. 5 Delivery of ZPAMt and decline in HIV-1 reservoir in the brain of HIV-1 infected hu-CD34 + NSG mice. a** Schematic, illustrating the experimental plan to check the efficacy of ZPAMt exosomes in controlling HIV-1 infection progression in hu-CD34 + NSG mice. **b** Dynamics of viremia in hu-CD34 + NSG mice after treatment with either nLuc exosomes (left), ZFP exosomes (center), ZPAMt exosomes (right) over the course of 10 weeks of infection and treatment as measured by RT-qPCR. Trendlines in red indicate the median of viral load. The pink line indicates LOD. **c** Whole-brain obtained after euthanizing HIV-1-infected hu-CD34 + NSG mice were subjected to formalin fixation and were paraffin-embedded. RNA was extracted from 10 micron thick slices from FFPE brain and RT-qPCR for Pol RNA using a Taqman probe was performed to determine the level of HIV-1 Pol mRNA in the brain of nLuc, ZFP, and ZPAMt exosome-treated mice. **d, e** RT-qPCR for Gag RNA from an equal amount of RNA from (**d**) bone marrow lysate (**e**) spleen tissue lysate from hu-CD34 + NSG mice of nLuc, ZFP, and ZPAMt exosomes treated mice. The line shows the median value in **c**–**e**. For **b**, **d** *n* = 7 were used for nLuc treated mice, *n* = 8 for ZFP treated mice and *n* = 10 for ZPAMt treated mice. For **b**, **d** Kruskal–Wallis test was performed to calculate *P* value, for **c**, **e** *P* value was determined by Kruskal–Wallis test followed by Dunn's post hoc test for multiple comparisons. Source data are provided in a source data file.

## Methods

**Plasmids**. Mammalian expression vector pcDNA3.1+ was used to express proteins ZFP362 ZFP362-DNMT3A or ZFP362-DNMT3A derived fusion constructs. For creating pcDNA-ZFP362-DNMT3A, the truncated and fusion constructs were ordered as gBlocks® (Integrated DNA Technology) and cloned into a pcDNA3.1(+) mammalian expression vector, digested with AfeI and Acc65I, using the NEBuilder® HiFi DNA Assembly Master Mix according to the manufacturer's instructions (New England Biolabs). EXOtic device plasmids pDB60 (booster), pDB68 (Connexin43 S368A), pSA465 (CD63-L7Ae) and pSA462 (nanoluc-C/D_box) was a kind gift from Dr Martin Fussenegger.[21] The C/D_box was introduced in the 3'UTR of ZFP362 and ZFP362-PAMt digested with Acc65I and NotI through

standard oligo cloning methods. The following reagent was obtained through the NIH HIV-1 Reagent Program, Division of AIDS, NIAID, NIH: Human Immunodeficiency Virus 1 (HIV-1-1) NL4-3 BaL Infectious Molecular Clone (p81A-4), ARP-11440, contributed by Dr. Bruce Chesebro[55]. The sequences of all gBlocks, oligos, and vectors are provided in Supplementary Data 1.

**Cell culture and transfection**. The HEK293T and immortalized Mesenchymal stem cells (MSC) were cultured in Dulbecco's modified Eagle's medium (DMEM) (Thermo Fisher Scientific) supplemented with 10% fetal bovine serum (FBS) (GeminiBio) and incubated at 37 °C and 5% $CO_2$. Immortalized MSCs expressing

human Telomerase Reverse Transcriptase with basal levels of GFP and Firefly luciferase were a kind gift from Dr. Glackin[31]. Jurkat cells were infected with 0.01 MOI of pNL4-3-derived HIV-1 to create chronically infected Jurkat cells[57]. The chronically HIV-1-infected Jurkat (CHI-Ju) were cultured in Roswell Park Memorial Institute (RPMI) 1640 (Thermo Fisher Scientific) with 10% FBS. CHI-Ju were electroporated using the 10 μL Neon™ transfection system (Thermo Fisher Scientific) as per manufacture's instruction under the following conditions: 1,325 V, 10 ms, 3 pulse. Peripheral Blood Monocytic Cells (PBMCs) were isolated using Histopaque®-1077 (Millipore Sigma) from donated blood and cultured in RPMI in the presence of IL-2 (100 units/ml) (Millipore Sigma catalogue number 11147528001) and 10% FBS. Microglial cells HC69.5 were a kind gift from Dr. David Alvarez-Carbonell, maintained in BrainPhys™ media (Stemcell technology catalogue number #05791)[58] and were electroporated using the 10 μL Neon™ transfection system (Thermo Fisher Scientific) as per manufacture's instruction under the following conditions: 1100 V, 50 ms, 1 pulse.

**Collection, purification and cellular exposure of exosomes**. In a 10 cm dish, $4 \times 10^6$ HEK293T cells were seeded and 24 h later transfected with 24.5 μg total of EXOtic device plasmids (14 μg of nLuc-C/$D_{box}$, or ZFP-C/$D_{box}$, or ZPAMt-C/$D_{box}$, and 3.5 μg each of the pDB60, pDB68 and pSA465 vectors). After 6 h of transfection, the cell culture medium was replaced with exosome-depleted FBS (Thermo Fisher Scientific). After 72 h of transfection, the media was collected and subjected to sequential spins: a low-level spin at $800 \times g$ for 5 mins to remove cells, followed by $2000 \times g$ for 20 mins to remove of cell debris and then subjected to ultra-centrifugation performed at $100,000 \times g$ for 2 h. The final precipitant was resuspended in sterile PBS and passed through a 0.22 micron Ultrafree® Centrifugal Filter Units (Millipore Sigma). The size distributions and an absolute number of exosomes were determined using the NanoSight NS300 (Malvern). For all in vitro experiments, an exosome to cell ratio of $10^4$ exosomes per cell was maintained.

**RT-qPCR from cells and exosomes**. RNA was isolated from CHI-Ju cells ($5 \times 10^5$) or exosomes ($1 \times 10^{10}$) using the Maxwell® automated simplyRNA isolation kit according to manufacturer instructions (Promega). Equal amounts of RNA were used for Luna® Universal One-Step RT-qPCR Kit (NEB) according to the manufacturer's instruction. Gene-specific primers sequences are listed in Supplementary Data 1. Roche® LightCycler 96 was used to perform the RT-qPCR and results were analyzed using the LightCycler 96 software (Roche). The PCR conditions were as follows: reverse transcription at 55 °C for 10 min; initial denaturation, 95 °C for 3 min; denaturation, 95 °C for 30 sec; annealing, 60 °C for 20 s for 35 cycles. For calculation of viral copy number/cell as done by Shan et al.[22], the equal number of cells were seeded and 100 ml of cell culture supernatant was harvested to isolate viral RNA using Maxwell® RSC Viral Total Nucleic Acid Purification Kit (Promega). These RNA samples along with RNA standard (AcroMetrix™ HIV-1 Panel) were subjected to RT-qPCR for the Gag region, the standard curve was plotted and viral RNA copy/ml was deduced.

**Negative staining transmission electron microscopy**. Exosomes were absorbed onto glow discharged carbon coated 200 mesh EM grids followed by conventional negative staining with 1% (w/v) uranyl acetate. Images were collected using an FEI Tecnai 12 transmission electron microscope (Thermo Fisher Scientific) equipped with a LaB6 filament and operated at an acceleration voltage of 120 kV. Images were recorded with a Gatan 2 k × 2 k CCD camera (Gatan, Inc.) using Gatan Microscopy Suite software GMS3 at a magnification of 30,000 × and a defocus value of ~1.5 μm[59].

**Western blotting**. Lysate from cells was prepared by resuspending the cells in Mammalian Protein Extraction Reagent (M-PER) (Thermo Fisher Scientific) buffer and protease inhibitor cocktail (Thermo Fisher Scientific) and subjecting them to sonication for 5 cycles of 30 s on and 30 s off. The same protocol was used for making lysates from exosomes suspended in 1% SDS containing RIPA buffer (50 mM Tris HCl, 150 mM NaCl, 1.0% (v/v) NP-40, 0.5% (w/v) Sodium Deoxycholate, 1.0 mM EDTA, 1% (w/v) SDS and 0.01% (w/v) sodium azide at a pH of 7.4). Protein samples were resolved on 10% SDS–PAGE gels (Bio-rad) and transferred onto PVDF membranes using Trans-Blot Turbo Transfer System (Bio-rad). The membranes were blocked with 5% non-fat milk (Millipore Sigma) in Tris-buffered saline (TBS) solution and incubated with primary antibodies in TBS with 0.05% Tween 20 (Millipore Sigma) (TBS-T) overnight at 4 °C. After washing, the blots were reacted with secondary antibodies for 45 min and developed using the enhanced chemiluminescence (ECL) detection system (Thermo Fisher Scientific). Anti-Myc antibody (Cell Signalling. catalogue no. 9B11), anti-Alix mouse monoclonal antibody (Cell Signalling, catalogue no. 3A9), anti-CD63 mouse monoclonal antibody (Santacruz Biotechnology, catalogue no. SC5275), anti-GAPDH mouse monoclonal antibody (Santacruz Biotechnology, catalogue no. SC47724), anti-Tsg101 mouse monoclonal antibody (Santacruz Biotechnology, catalogue no. SC7964), anti-tubulin rabbit monoclonal antibody (catalogue no. abcam, ab6046). All primary antibodies were used at a dilution of 1:1000. Goat Anti-Mouse IgG (H + L)-HRP Conjugate (Biorad catalogue no. #1706516) or Goat Anti-Rabbit IgG (H + L)-HRP Conjugate (Biorad catalogue no. #1706515) was used at 1:4000 dilution in 1% non-fat milk (Millipore Sigma) in TBS-T.

**Chromatin Immunoprecipitation (ChIP) and Methyl DNA precipitation assay (meDIP)**. For ChIP or meDIP assay, after four days of transfection with empty vector, ZFP, or ZFP fusion constructs in CHI-Ju cells, $2.5 \times 10^6$ cells were harvested for each sample and subjected to the protocol as described previously[60]. Briefly, cells were cross-linked, quenched, and lysed. The cell lysate was sonicated with a Bioruptor (Diagenode) using pulse interval 30 s ON and 30 s OFF for a total of 30 cycles. For the ChIP assay, 5μl Pierce™ Anti-c-Myc Magnetic Beads (Thermo Fisher Scientific) per $2.5 \times 10^6$ cells, were added to the mixture and after incubation at 4 °C for 2 h, beads were washed with a series of buffers with increasing salt concentrations. Finally, DNA was eluted and reverse cross-linked and purified with QIAquick PCR Purification Kit (Qiagen) and the relative enrichment of Myc tagged ZFP or ZFP fusion constructs was determined at the HIV-1 LTR using qPCR. Primer sequences are listed in Supplementary Data 1.

For the MeDIP assay, CHI-Ju cells were harvested 10 days after transfection and subjected to the same treatment as above until after the sonication step. An antibody against 5-methylcytosine (5-mC) (Abcam catalogue no. ab10805) or non-specific IgG was added to the cross-linked and sonicated lysates (1 mg antibody per $10^7$ cells). After antibody incubation overnight, ChIP-grade Protein A/G magnetic beads (Thermo Fisher Scientific) were added for 2 h. Beads were washed in low and high salt buffers[60] and proceeded as per ChIP protocol.

After ChIP or meDIP, DNA was subjected to qPCR assay to calculate fold enrichment using Luna® Universal qPCR Master Mix (NEB). Ct values for input samples were adjusted for dilution and fold enrichment was calculated as 100*2^(Adjusted input - Ct (IP)).

**Lentiviral plasmid construction and lentivirus production**. The exosome producer third-generation Lentivirus plasmid was custom built by VectorBuilder. Inc., which contained a hPGK1 promoter driving expression of CD63-L7Ae and Connexin43 S368A and EF1α promoter driving ZFP362-PAM- C/$D_{box}$ expression. The inclusion of protease cleavage site P2A between CD63-L7Ae and Connexin43 S368A ensured protein separation. Mutations in the bovine growth hormone polyadenylation (bgh-PolyA) transcription termination signal after CD63-L7Ae and Connexin43 S368A enable read-through and expression of complete lentiviral genomic RNA expressed from the RSV promoter. The faulty poly (A) signal allowed for CD63-L7Ae and Connexin43 S368A expression. To obtain pLV-EXOtic-ZFP, DNA fragment between Afe1 and Acc65I i.e., PAMt was removed, overhangs filled in using Klenow Fragment (NEB) and re-ligated. To build the pLV-EXOtic-nLuc vector, the pLV-EXOtic-ZPAMt vector was digested with NsiI and BstBI (NEB) to remove the ZPAMt and replaced with the *nLuc* gene using NEBuilder HiFi DNA Assembly Master Mix (NEB) according to the manufacturer's instructions. The calcium phosphate method (Takara) was used to transfected HEK 293 T cells with a mixture of genome transfer plasmid and packaging plasmid: pRRE, pCMV.Rev, pMD2.G[61], at a ratio of 4:2:1:6[62]. The lentivirus vector was used at an MOI of 5 to transduce Mesenchymal Stem cells via spinoculation.

**Mice**. NOD.Cg-Prkdcscid Il2rgtm1Wjl/SzJ (NSG) mice (JAX stock #005557, Jackson Laboratory) were maintained in accordance with the Guide for the Care and Use of Laboratory Animals and were housed in Specific Pathogen Free conditions. Animal housing follows a 12 h light/12 h dark cycle, wherein the temperature is maintained between 68–79 ˚F and humidity was maintained between 30–70 percent. All experiments were performed according to the guidelines of the Institutional Animal Committee of the Beckman Research Institute of the City of Hope.

**Exosome biodistribution**. Nanoluc exosomes were administered into NSG mice in a total of 100 μl volume via RO injection. Four hours after exosome injection, mice were euthanized using an overdose of isoflurane anesthesia. Brain, bone marrow, liver, and spleen were harvested in PBS with a 1X protease inhibitor cocktail (Thermo Fisher Scientific). Organs were homogenized and rinsed with ACK lysis buffer (Thermo Fisher Scientific) before resuspending them in 1 ml of PBS. Nano-Glo® Luciferase assay system (Promega) was used to determine the luciferase activity in the organ lysates as per the manufacturer's instructions. One hundred microliters of lysate were added to 100 μl of Nano-Glo® luciferase assay buffer and mixed well before adding 1:25 prediluted Nano-Glo® luciferase substrate and luciferase activity detected on the Promega GloMax Discover Microplate Reader Detection System with GM3000 Software (Promega).

**Transplantation of HIV-1 infected PBMC in NOD/SCID/IL2rnull mice (hu-NSG)**. Development of the hu-PBMC NSG mouse model was described in[63]. PBMC were cultured in RPMI containing 10% heat-inactivated FBS and IL-2 (100 U/ml) and activated overnight using 25 μl/ml CD3CD28 activator (Stemcell ImmunoCult #10990). Activated PBMCs were infected with 200 ng p24 equivalent of pNL4-3 BAL derived HIV-1 in the presence of 2 μg/ml polybrene. After two days of infection, $1 \times 10^6$ activated, infected PBMCs were mixed with $9 \times 10^6$ uninfected PBMCs. A total of $1 \times 10^7$ PBMCs resuspended in 200 μl of sterile PBS were injected IP into 6 to 8-week old NSG mice (NOD.Cg-Prkdc scid Il2rg tm1Wj/SzJ (NOD/SCID/ IL2rnull [NSG]) obtained from Jackson Laboratories. Mice were bled by retro-orbital bleeding and plasma viral loads were analyzed periodically RT-

qPCR as described later. Procedures involving HIV-1 or HIV-1-infected mice were performed under general anesthesia.

**Establishment of hu-CD34+NSG Model.** Engraftment of purified human CD45 + HSC in NSG pups and their HIV-1 infection was done as described elsewhere[35]. For preparation and engraftment of hu-CD34 cells, human CD34 + HSCs were isolated from fetal liver and thymus tissue obtained from Advance Bioscience Resources following regulatory guidelines. The fetal liver tissue (16–24 weeks of gestation) was treated with collagenase digestion, followed by filtration through a sterile 70 μm nylon mesh filter. Immunomagnetic enrichment for CD34 + cells were performed using EasySep™ Human Cord Blood CD34 Positive Selection Kit (Catalog #18096) (Stem Cell Technologies, Vancouver, BC), per the manufacturer's instructions. This kit gave us >90% pure CD34 + population, as measured by flow cytometry analysis using an anti-CD34 antibody. For engraftment, a modified intrahepatic injection technique was used for the engraftment of neonatal pups within 72 h of birth. A custom-made Hamilton 80508 syringe/needle was used for the injections (30-gauge, 15 mm long needle with a bevelled edge attached to a 50 μL glass syringe). The maximum volume used for injection with this needle/syringe was 30 μL per pup. Animals were pre-irradiated with 100 cGy 137Cs then allowed to recover in the home cage with heat support for 4–6 h before transplantation. Pups were anesthetized by hypothermia anesthesia, then transplanted with $1.0 \times 10^6$ CD34 + cells each by intrahepatic injection. 10–12 weeks after transplantation, blood was collected, and the engraftment was verified using multi-parameter flow cytometry analysis.

**Antiretroviral therapy and exosome dose administration.** Infected mice with detectable viremia were treated orally for two weeks with cART composed of drugs that block new infections. The cART regimen consisting of Truvada® [tenofovir disoproxil fumarate (TDF; 300 mg/tablet), emtricitabine (FTC; 200 mg/tablet) (Gilead Sciences)] and Isentress® [raltegravir (RAL; 400 mg/tablet) (Merck)], scaled down to the equivalent mouse dosage using the appropriate conversion factor, was administered in a drinking water formulation (sweetened water gel, Medidrop® Sucralose, ClearH$_2$0). For 200 ml Medidrop®, ½ Truvada tablet, and ½ Isentress tablet were crushed to powder, mixed by vigorous shaking, and changed weekly as per doses calculated previously[64]. Mice were bled by retro-orbital bleeding, and peripheral blood cell populations and plasma viral loads were analyzed periodically using RT-qPCR. The oral cART regimen was withdrawn after two weeks and exosome treatment was initiated. Exosomes derived from HEK293T cells packed with nLuc, ZFP, or ZPAMt were retro-orbitally injected at a concentration of $100 \times 10^9$ in 100 ul sterile PBS once a week for 6 weeks.

**Viral load calculation using RT-qPCR.** Serum obtained from retro-orbitally collected blood was subjected to automated RNA isolation using Maxwell® RSC Viral Total Nucleic Acid Purification Kit according to the manufacturer's instructions (Promega). Viremia was assayed using TaqMan® Fast Virus one-step reverse transcriptase real-time PCR (Thermo Fisher Scientific) with an automated CFX96 Touch Real-Time PCR detection system (Bio-Rad). Copies/ml of HIV-1 RNA was calculated using AcroMetrix™ HIV-1 Panel copies/mL (Thermo Fisher Scientific #950470). The LODs of the mice plasma were found to be around 200 RNA copies/ml under these experimental conditions. The mice were bled and assayed for the rebound of plasma viremia and peripheral cell composition periodically until euthanized. Before euthanizing, under anaesthesia, mice were bled by cardiac puncture to collect 700 μl of blood. After euthanizing, spleen and bone marrow (BM) were harvested from all the mice. Lysates from the spleen and BM were treated with ACK lysis buffer (Thermo Fisher Scientific) and washed with PBS before adding homogenization buffer from Maxwell® RSC simply RNA Kits (Promega) to $1 \times 10^6$ cells. RNA was isolated and HIV-1 Gag RNA expression was measured from equal amounts of RNA using Luna® Universal One-Step RT-qPCR Kit (NEB) using HIV-1 Gag region and β-Actin specific primers. From the formalin-fixed paraffin-embedded brain tissues of hu-CD34 + NSG mice, RNA was extracted using Maxwell® RSC RNA FFPE Kit (Promega) as per the manufacturer's protocol. Taqman probes and primers used for Gag region and bActin PCR are listed in Supplementary Data 1.

**MeDIP-qPCR from mice samples.** MeDIP from bone marrow samples from all the HIV-1-infected huNSG-PBMC mice was carried out using the EpiQuik MeDIP Kit (Epigentek) as per the manufacturer's instructions. This kit included a ChIP-grade anti-5-methylCytosine (5mC) antibody and normal mouse IgG for negative control. Briefly, DNA from bone marrow was extracted and sonicated using Bioruptor (Diagenode) for 5 cycles using pulse interval 30 s ON and 30 s OFF, shearing into 200–1000 bp fragments. An aliquot of each sample was set aside as input control, while the remaining portion was subjected to immuno-precipitate with an anti-5mC antibody or normal mouse IgG (1 μg per $2.5 \times 10^5$ cells) provided in the kit. DNA was released from the antibody-DNA complex by proteinase K and purified through the specifically designed Fast-Spin Column provided in the kit. Eluted DNA was amplified with primers specific for HIV-1 LTR encompassing ZFP binding site and H2B1A (Supplementary Data 1) with PCR conditions as above.

**Toxicity assays.** After 10 weeks of infection and therapy regimen, mice were euthanized, and total blood was harvested. Blood was spun at 10 min at $1000–2000 \times g$ using a refrigerated centrifuge to collect plasma and 20 μl of plasma from each mouse in duplicate was used to determine ALT and AST activity using the Alanine Transaminase Colorimetric Activity Assay Kit or Aspartate Aminotransferase Colorimetric Activity Assay Kit according to the manufacturer's instructions (Cayman Chemical).

**Flow cytometry.** Samples of peripheral blood were collected by retro-orbital bleeding under general anesthesia and stained for 30 min with 2 μl BUV395-conjugated anti-human CD45 antibody (BD Biosciences, BDB563792), 2 μl BV711-conjugated anti-human CD3 antibody (BD Biosciences, BDB563725), and 5 μl APC-conjugated anti-human CD4 antibody (BD Biosciences, BDB555349). (Stained peripheral blood samples were then lysed with red blood cell lysis buffer and absolute cell counts calculated using BD Liquid Counting Beads (BD Biosciences 335925,). Flow cytometry was performed using a BD Fortessa II instrument (BD Biosciences) and analyzed with the FlowJo software (BD Biosciences).

**Immunohistochemistry (IHC).** Mouse brains were fixed in 10% formalin post-necropsy. Fixed brain samples were processed by the Veterinary Pathology Core (City of Hope) into three coronal brain regions, namely levels I (Bregma 1.7-0.14), II (Bregma −2.18 to −2.8), and III (Bregma −5.27 to −7.08). Ten million chronically infected Jurkat cells were washed 2X with PBS, before fixing in 10% formalin for 10 mins. Cells were processed into a paraffin-embedded cell block. Processed and fixed samples were subsequently used for immunohistochemistry (IHC). IHC was performed on Ventana Discovery Ultra IHC automated Stainer (Ventana Medical Systems, Roche Diagnostics). Briefly, formalin-fixed paraffin-embedded tissue blocks were sectioned at a thickness of 5 μm and put on positively charged glass slides. The slides were loaded onto the machine; deparaffinized, rehydrated, endogenous peroxidase activity inhibited, and subjected to antigen retrieval. The slides were then incubated with a primary antibody followed by DISCOVERY anti-Rabbit HQ and then subjected to a DISCOVERY anti-HQ HRP detection system. The stains were visualized by a DISCOVERY ChromoMap DAB Kit, and counterstained with hematoxylin (Ventana), and covered with a coverslip. The primary antibody used was an HIV-1 p24 (NBP2-89962; Bio-Techne, clone 002) diluted 1:2000 for IHC. Slides were scanned by the Light Microscopy Core (City of Hope) on a Hamamatsu Slide Scanner at a 20 X magnification. To analyze the IHC slides, we used QuPath version 0.2.24 (The University of Edinburgh, UK).

**Ethics statement.** All animal care and procedures have been performed according to protocols reviewed and approved by the City of Hope Institutional Animal Care and Use Committee (IACUC) held by the principal investigator for this application (John Burnett and Kevin Morris, IACUC 16095). The research involves blood specimens from anonymous human subjects with no Health Insurance Portability and Accountability Act of 1996 (HIPAA) identifiers. Discarded peripheral blood from anonymous, healthy adult donors from the City of Hope Blood Donor Center (Duarte, CA) were used for the isolation of PBMCs under a protocol determined by the City of Hope Institutional Review Board (IRB number 19582) to be Not Human Subjects Research under OHRP's Coded Specimen Guidance. Donors have given their informed consent on the utilization of blood for research purposes on the Blood Donor Center's Donor Consent Card. Foetal tissue and cord blood units were obtained under a protocol determined by the City of Hope Institutional Review Board (IRB number 17155) to be Not Human Subjects Research under OHRP's Coded Specimen Guidance and therefore did not require informed consent. These specimens were obtained from commercial specimen vendors with no HIPAA identifiers. Both projects were evaluated by the Institutional Review Board (IRB) of the City of Hope and determined not to involve human subjects research per federal regulation 45 CFR 46.102 (d)(f); IRB#/REF#: 19582/184924 and 17155/141478 and comply with the declaration of Helsinki.

**Statistical analysis.** Statistical analysis was performed with GraphPad Prism 8.3 software. Data are shown as mean ± SD of triplicates unless otherwise indicated. Statistical analysis was performed using a two-tailed Student's t-test or one-way ANOVA with post hoc tests, as described. A p value less than 0.05 was designated as statistically significant. No animals were excluded from the analysis.

**Reporting summary.** Further information on research design is available in the Nature Research Reporting Summary linked to this article.

## Data availability
The source data underlying all Figs and Supplementary Figs. 1–9 and 12 are provided as a Source Data file. All other relevant data supporting the key findings of this study are available within the article and its supplementary files or from the corresponding author upon reasonable request. Source data are provided with this paper.

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

## Acknowledgements

We thank Dr. Zhuo Li, and Ricardo Zerda at the City of Hope Electron Microscopy Core Facility. Research reported in this publication included work performed in the Veterinary pathology core, the Light Microscopy Core, and the Anatomic Pathology Core at the City of Hope. We would like to thank Martha Salas and Dr. Brian Armstrong for their help with the slide processing. We would also like to thank Lori Sagnek of the veterinary pathology core for processing and sectioning the mouse brain samples and Dr. Aimin Li and her team at the anatomic pathology core for processing the sections for immunohistochemistry. We thank Servier Medical Art (http://smart.servier.com/) for providing medical art images for illustration. This work was supported by NIMH R01 113407-01 to KVM.

## Author contributions

S.S., T.S. and K.V.M. conceived and designed the experiments. S.S., R.M.R. and N.A.G. performed the experiments. S.S., L.E., L.H., R.M.R. and J.B. designed and conducted animal experiments. S.S, L.H. and R.M.R. analyzed the data and prepared the figures. S.S., T.A.S. and K.V.M. wrote the paper. All authors have reviewed and edited the paper.

## Competing interests

Development and characterization of ZFP-362 has been filed for a patent under international application number PCT/US18/56943 by KVM and TAS. The remaining authors declare no competing interests.
