## [Peer Review File · Nature Communications]

REVIEWER COMMENTS

Reviewer #1 (Remarks to the Author):

Shrivastava et al have developed a zinc finger protein targeting the HIV-1 LTR and demonstrate that exosomes packaged with this ZFP-362 fused to DNA methyltransferase 3A domains can repress the integrated HIV-1 provirus. The work is interesting, and conducts in vivo proof of principle studies in humanized mice to advance this intervention as a potential 'block and lock' approach for stably silencing HIV-1 in larger animal models or humans. It could be of interest to Nature Communication readers.

Major concerns/critiques-

- The authors provide no information whatsoever about the development and characterization of ZFP-362, besides the information that it targets the NFκB doublet in the HIV-1 LTR. It is unclear how this was identified or evolved and what the off-target effects are for this ZFP in terms of binding the multiple other NFκB responsive promoter elements that are present in the human genome. This is particularly of concern as ZFP-362, by itself appears to enhance transcription of the LTR in several experiments presented in the manuscript (see below).
- ZFP362 by itself enhances gag expression in Fig. 1B, 1H, 3G, S1D
- In Fig. 2H, it is unclear why there is no increase in gag RNA expression in the control cells (black bars) and cells transfected with ZFP alone in response to all inducers (except TSA) in comparison to the vehicle control. TNF-α is a strong inducer in all standard latent cell line model of HIV-1, and even this does not seem to elicit gag RNA expression in control cells and 'ZFP alone' samples. It would perhaps be better to represent these as pre- and post- induction samples after treatment with the inducers.
- It is unclear if the exosomes preparations are pure enough to exclude microvesicles. Also unclear how the ZFP fusion protein derived from the exosomes localizes to the nucleus.
- Most data is represented as fold changes in gag RNA expression which is misleading. This is misleading and in fact very modest (about 2 fold in the best case) when considering experiments in cell lines and primary cells. It would be better represented as the number of copies per million cells. The same suggestion applies to data I animal experiments.
- Fig S3: The nLuc activity appears to vary widely across experiments despite using similar dosage- the normalized data recorded with 140 billion exosomes in S3b appears identical to that obtained with 20 billion exosomes in S3a. Why is this so? Along the same lines, published literature demonstrate that retro-orbital route of exosome injection (refer to studies from Paul Crocker's group) leads to trafficking through the heart and liver and accumulation in the spleen. Is there a reason that the exosomes here appear to localize more the bone marrow and the brain tissue?
- The Hu-PBMC-NSG model is a short term acute model for HIV-1 infection, which very rapidly displays CD4 T cell loss with increasing in HIV VLs within a couple of weeks of established infection. The 1 week period of ART treatment appears to have no effect whatsoever on the plasma VLs in these mice. In the absence of a CD4 T cell profile, it is difficult to gauge if VL control is because of reduced CD4 T cell frequencies. It is also unclear, what is the HIV-infected cell type targeted in the BM in these animals by the exosomes and what the levels of infection and impact of treatment are in hematopoietic tissues like the spleen which are a major HIV reservoir in the Hu-PBL model.
- Very similar concerns with the experiments in the hu-CD34+ NSG mice model. This is not an optimal model for the CNS reservoir. ART treatment does not appear to suppress the VLs and in the absence of CD4 T cell data, it is unclear if the effects observed are due to CD4 T cell decline rather than the treatment intervention. In what cell types are the exosomes suppressing the provirus? What are the levels of viral RNA in the spleen and gut tissue?

Reviewer #2 (Remarks to the Author):

The manuscript by Surya Shrivastava and colleagues titled, "Exosome-mediated stable epigenetic repression of HIV-1" describes an in-depth study whereas virus infected cells are targeted for suppression by epigenetic means to induce a stable "block and lock" inhibition for viral-gene expression. An HIV-1 promoter was developed to target a Zinc Finger Protein (ZFP-362) shown to bind to an NFκB doublet on the HIV-1 LTR. ZFP-362 was fused to a spectrum of domains of DNA methyltransferase 3A to improve viral suppression efficiency. Exosomes were engineered to package and deliver this therapeutic cargo. Results were affirmed in humanized mouse models of HIV-1 infection. The works serve to suggest that exosome-mediated systemic delivery platform for therapeutic cargos to epigenetically repress HIV-1 infection can provide a novel therapeutic directive to combat ongoing HIV-1 infection/replication.

While the results are of potential interest there are a number of major concerns for this work in its current form.

First, the specificity of ZPAMt to stably represses HIV-1 in a long term and specific manner by targeting CpG methylation of the 5' LTR warrants clarifications. The fact that the repression is relatively resistant to reactivation is shown in one plate of a single experiment with viral measures at the transcriptional level. These results should be affirmed over daily time periods with measures of progeny virus (reverse transcriptase activity) production, viral protein synthesis along with viral RNA measures. The absolute differences in the levels of virus being induced need be shown. These cross validating experiments are important in that it forms the basis for why ZPAMt was selected and used to make it deliverable to virus infected cells through exosomes. Re-analysis of the virological tests should include additional biological replicates and positive controls beyond what is now illustrated for ZFP-362. These additional assays will affirm these critical data sets.

Second, while the exosome system used to deliver mRNA for the epigenetic inhibitor of HIV-1 report that they can induce an epigenetic "block and lock" of HIV-1 expression the results given are not uniform in that different donor PBMC show variant results. These data sets should be affirmed with multiple viral strains given at different multiplicities over varied time points and beyond viral RNA for measures of actual virus production through RT activity measures in culture fluids. The data may also be replicated by using a transformed cell line in measurements.

Third, while the observations purport to provide proof of principle demonstration for the utility of exosomes for epigenetics-based therapeutics in vivo the results seen in humanized mice and for both PBMC and CD34 reconstituted mice are highly variable. An explanation should be provided for the different results seen and the variant experimental protocols used. A secondary concern is the transient use of antiretroviral therapy for one to two weeks without measured drug levels and evidence of complete viral suppression. This questions the results for the conclusions of stable targeted modulation of gene expression. Differences are observed in bone marrow and spleen which are not easily understood but need be explained.

Fourth, the data set illustrated for the brain is difficult to interpret as there is no reconstitution of human cells in the brain and the only cells seen in these animals are cells that migrate from the periphery to the CNS. No primary neural cells are of human origin. There again needs to be some explanation as to what is being recorded in the brain experiments.

Fifth, in order to achieve stable transcriptional repression of HIV-1 the authors developed a vector expressing recombinant fusion protein, ZFP 362b-DNMT3A (ZD3A) this was reported as being capable of binding to the NF-κB doublet binding site within LTR promoter of HIV-1 and fused to a DNA methyl transferase 3 domain (3A) to generate. Once again a more complete description of these constructs and its development is in order.

Sixth, I do not see how a transfection assay can be monitored for 10 days? What is the evidence for gene expression to continue this long? The transfection of ZD3A into HIV-1 infected Jurkat cells (CHI-Ju) is noted but the levels of viral production (for example, RT activity in culture supernatant fluids) was not shown. The reduced HIV-1 expression should be shown over time by more than one viral detection system. The TNF induction was confusing as why was TNF needed if the cells are continuously releasing virus?

Was another cell line used? If so how was it characterized? Was PMA, LPS or other inducers used to affirm the test results? The TNF appears to be a single time point and as such was this affirmed over time and if so what time points were measured? Why was a model of HIV-1 infection of microglial cells used as in none of the humanized mice are microglial cells of human origin present?

Reviewer #3 (Remarks to the Author):

The manuscript by Shrivastava et al., provided an interesting approach to suppress the replication of integrated HIV provirus. Epigenetic modulation with selected fusion protein delivered by exosomes also is an attractive approach.

However, it is not as practical as of today. Delivery of foreign proteins will induce immune responses and minimize therapeutic efficacy.

There are some technical problems with in vivo studies on humanized mice.

The suppression of HIV replication by multiple injections of exosomes with selected construct is not convincing.

There are some problems with the results.

Figure 4. One week of ART treatment not clear what for and not explained in the text. The viral load declined spontaneously in control exosomes treated mice.

Induction of latency (no production of viral particles and new infection of available naïve CD4+ cells) in vivo in the presence of actively replicating virus is very questionable. Sometimes in hu-PBL mice with time virus destroy available CD4+ cells and viral load drop as no more targets for active replication and viral production.

Information about the absolute number of human CD4+ cells is important and needs to be included. To confirm the reduction of HIV replication all HIVgag PCR values should be adjusted to the number of human cells in both models.

Figure 5. The ART treatment did not result in suppression of HIV replication. The strange result with an increased viral load from 2 to 4 weeks post-infection.

In original reference #52, which was used as a protocol for the treatment of mice "Viremic hu-NSG mice exhibited a robust suppression of viral load in response to the oral three-drug cART regimen within 2 weeks of treatment" with significant recovery of CD4+ cells.

Material and Methods need to provide information about the infection of humanized NSG mice. Reference #52 has at least three HIV strains. Which one and at what dose did the authors use?

Minor mistake:

Reference 19 is not Kojima et al. It is #27.

Response to Reviewers comments

We would like to thank the reviewer for their valuable and productive comments on this manuscript. Our comments in response to the reviewer comments are shown below in *italics* and underlined for clarity.

REVIEWER COMMENTS

Reviewer #1 (Remarks to the Author):

Shrivastava et al have developed a zinc finger protein targeting the HIV-1 LTR and demonstrate that exosomes packaged with this ZFP-362 fused to DNA methyltransferase 3A domains can repress the integrated HIV-1 provirus. The work is interesting, and conducts in vivo proof of principle studies in humanized mice to advance this intervention as a potential 'block and lock' approach for stably silencing HIV-1 in larger animal models or humans. It could be of interest to Nature Communication readers.

Response: Thank you for recognizing the impact of this work and helping us to clearly explain those experiments which were carried out to determine if the approach outlined here could prove worthwhile as a means to delivering anti-HIV therapeutic modalities to infected cells in vivo.

Major concerns/critiques-

(1) The authors provide no information whatsoever about the development and characterization of ZFP-362, besides the information that it targets the NFκB doublet in the HIV-1 LTR. It is unclear how this was identified or evolved and what the off-target effects are for this ZFP in terms of binding the multiple other NFκB responsive promoter elements that are present in the human genome. This is particularly of concern as ZFP-362, by itself appears to enhance transcription of the LTR in several experiments presented in the manuscript. ZFP-362 by itself enhances gag expression in Fig. 1B, 1H, 3G, S1D.

Response: A recent paper published from our group goes into the details of ZFP-362 functional interactions and ability to target the NF-κB doublet which is unique to the HIV LTR and not found elsewhere in the human genome¹. We have added this citation (PMID: 33335944) to the resubmission. As per the reviewer's suggestion we have also added a brief description of ZFP-362 to the manuscript. The observed enhancement of Gag expression is not significant in Fig 1B, and is variable as observed in 3 out of 5 PBMC donors (Fig 3G). In long term repression studies, we first see a decline in gag RNA expression due to ZFP and then an insignificant increase in later time points (Figure 2G). The inherent variability in effect of ZFP permits D3A or PAMt to assert a dominant repressive effect at the bound DNA. However, we cannot rule out that ZFP-362 binding to this NF-κB doublet in the HIV LTR is having an effect on chromatin at the LTR, but this was not the focus of this study. However, if this were the case then it

would be a useful feature of ZFP-362 fusions, as it would permit accessibility of proteins to the DNA and enhance the ability to epigenetically modify the 362 target site in the LTR.

(2) In Fig. 2H, it is unclear why there is no increase in gag RNA expression in the control cells (black bars) and cells transfected with ZFP alone in response to all inducers (except TSA) in comparison to the vehicle control. TNF- α is a strong inducer in all standard latent cell line model of HIV-1, and even this does not seem to elicit gag RNA expression in control cells and 'ZFP alone' samples. It would perhaps be better to represent these as pre- and post- induction samples after treatment with the inducers.

Response: The cell lines used for these set of experiments are the chronically HIV-1 infected Jurkat cells which have persistent infection, which is not latent, but remains a low-level infection. Since they are not latent, they do not get activated with TNF α or Azacytidine treatments and thus we did not observe any increase in Gag expression (refer to the black bars Fig 2H). Surprisingly we see an increase in Gag mRNA transcription under the influence of TSA. TSA being an HDAC inhibitor can influence HIV-1 transcription in ways other than activating latency. For instance, TSA has been observed to promote acetylation of HIV-1 trans activator (Tat) protein which then drives HIV-1 transcription². Furthermore, we cannot present a set of vehicle-treated cells as pre- or post-induction because all cells with small molecule treatment or vehicle treatment were being monitored simultaneously.

(3) It is unclear if the exosomes preparations are pure enough to exclude micro vesicles. Also unclear how the ZFP fusion protein derived from the exosomes localizes to the nucleus.

Response: This is an excellent point. Regrettably, there are no markers which confidently differentiate between exosomes and micro-vesicles³. However micro vesicles (150 to 1000 nm) and exosomes (50 to 150 nm) differ in terms of size. Characterization via nanosight tracking analysis and transmission electron microscopy (Figure 3B) confirms that the nanoparticles we were handling are classified under the exosome category of extracellular vesicles. Notably all control and treatment exosome are isolated in the same manner so if there is a contribution of microvesicles in the various experiments it would be presumed to be equally impacting in the control and treated samples. ZFP fusion protein has nuclear localization signals (NLS) i.e. Tat NLS⁴ and nucleoplasmin NLS⁵ as is also depicted in Figure 1A which facilitates the ZFP localization to the nucleus. After the exosomal cargo is delivered to cytoplasm in the recipient cells, exosome bound ZPAMt mRNA can express ZPAMt protein which localizes to nucleus with the help of the NLS.

-(4) Most data are represented as fold changes in gag RNA expression which is misleading. This is misleading and in fact very modest (about 2-fold in the best case) when considering experiments in cell lines and primary cells. It would be better represented as the number of copies per million cells. The same suggestion applies to data I animal experiments.

Response: We chose to present the data as fold change in gag RNA expression as this is conventionally done and intuitive for the readers to easier interpretation. As per reviewer's suggestion viral copy number/cell data have now been added in Supplementary Fig.1a and b which notably corroborates with the Gag RNA expression data. As the viral copy/ml data and p24 assay data are in sync with ZD3A mediated repression of Gag RNA data in Fig. 1D we found it redundant to repeat these assays for every in vitro experiment. For the animal experiments, RNA was extracted from lysates of tissue which is a heterogenous population of murine and human cells, making it difficult of figure out the number of cells, as the copy number/cell will not be reflective of infected cells. As such we reasoned that fold change is the ideal manner with which to present the data as it allows for in vitro data to be contrasted more easily with the in vivo data.

(5) Fig S3: The nLuc activity appears to vary widely across experiments despite using similar dosage- the normalized data recorded with 140 billion exosomes in S3b appears identical to that obtained with 20 billion exosomes in S3a. Why is this so? Along the same lines, published literature demonstrate that retro-orbital route of exosome injection (refer to studies from Paul Crocker's group) leads to trafficking through the heart and liver and accumulation in the spleen. Is there a reason that the exosomes here appear to localize more the bone marrow and the brain tissue?

Response: The variation in observation can be attributed to inter assay variability depending on age of mice, differences in handling and other technical aspects. Nonetheless we learned that the Retro Orbital (RO) route was most convenient for exosomes to access the blood and the brain, bone marrow and spleen, which was consistently observed across both experiments (Figure S3a and b). As the reviewer pointed out exosome delivery to spleen via intravenous route is also shown in ⁶. The biodistribution study data presented here allowed for a comparison of inter organ accumulation of exosomes based on Nanoluc activity as a means for us to gauge which route would be most advantageous for the in vivo injection of exosomes. Nanoluc activity provides a good indication of successful uptake of engineered exosomes into these organs, as the nanoluc mRNA needs to be expressed in order to see signal. In the absence of inter-tissue normalization, we do not claim organ specific preference of engineered exosome uptake. However, this is no doubt a highly interesting field of study and the source of much ongoing experimentation in the laboratory, e.g. to direct exosomes to target cells. Paul Crocker's group pursued such an effect by deletion of CD169 which resulted in retention of florescent labelled splenocyte derived exosomes within spleen following IV injection ⁶. In this study they did not check for exosome accumulation in the bone marrow or brain and/or tested the functionality of unloaded cargo. Although differing methodologies, we share commonality in the findings that the exosomes are lost from systemic circulation within a few hours following intravenous injection.

(6) The Hu-PBMC-NSG model is a short-term acute model for HIV-1 infection, which very rapidly displays CD4 T cell loss with increasing in HIV VLs within a couple of

weeks of established infection. The 1 week period of ART treatment appears to have no effect whatsoever on the plasma VLs in these mice. In the absence of a CD4 T cell profile, it is difficult to gauge if VL control is because of reduced CD4 T cell frequencies. It is also unclear, what is the HIV-infected cell type targeted in the BM in these animals by the exosomes and what the levels of infection and impact of treatment are in hematopoietic tissues like the spleen which are a major HIV reservoir in the Hu-PBL model.

Response: We agree with the reviewer's sentiments here on in vivo models of HIV infection. We have provided data in which we observed cART mediated control of HIV (Supplementary Fig. 6). We also observed that the no cART-treated hu-PBMC mice after two weeks of infection showed ~3 to 4 log higher viremia than the cART treated (Supplementary Fig. 6). With the short term oral cART treatment, a decline in infection was observed in 10 out of 18 mice at week 4 compared to week 2 (Figure 4b). This observed individual variability in mice may be attributed to a technical issue with the dosing of cART which is carried out through the drinking water. As such the two weeks of oral ART may not be exhaustive but still retain some moderate effect in controlling HIV infection

As the reviewer has rightly pointed out that the hu-PBMC NSG mouse model presents rapid loss of CD4+ cells, and we have added data showing the relative increase in CD4+ T-cells during ART treatment (Supplementary Fig. 6). Furthermore, as per the reviewer's suggestion, CD4+ rescue data from hu-PBMC mice has now been added in Supplementary Fig. 12 b. Surprisingly, we observed an increase in CD4+ cells in all the three groups suggesting exosomes by themselves have CD4+ rescue effects and may be explained by increased proliferation, activation or differentiation of CD4+ cells by exosome treatment which has also been extensively established in literature⁷⁻¹⁷. Varying in methodology and outcome all these studies have a common underlying factor that exosomes, irrespective of source, boost CD4+ T-cells. As only ZPAMt packed exosomes can exert HIV-1 suppressing effects, we observed significant repression in viremia levels (Figure 4B) at end of week 10. To confirm that ZPAMt mediated repression of viremia is not a result of the loss of CD4+ cells, we baselined viremia by absolute CD4+ cells in each mouse and have now presented these data in Supplementary Fig. 12d.

As per reviewer's remark on cell type targeting in BM, we know that exosomes are efficiently taken up by CD4+ cells and PBMCs. We observed with in vitro studies that Jurkat cells and PBMCs take up HEK293T/MSC derived exosomes and show significant increase in Nanoluc activity (Supplementary Figure 4). We did observe a strong trend in exosomal ZPAMt mediated HIV-1 transcription suppression in hematopoietic organs like bone marrow and spleen from hu-PBMC NSG mice cohort (Supplementary Fig. 7). While interesting, the study of individual cell types, the extent of exosome uptake and response to therapeutic packed exosome in various organs and their enrichment would be a separate study altogether and is outside the scope of this current body of work.

Very similar concerns with the experiments in the hu-CD34+ NSG mice model. This is not an optimal model for the CNS reservoir. ART treatment does not appear to suppress the VLs and in the absence of CD4 T cell data, it is unclear if the effects observed are due to CD4+ T cell decline rather than the treatment intervention. In what cell types are the exosomes suppressing the provirus?

Response. We appreciate the reviewer's comment, but all the recent reviews on animal models of HIV-1 progression in brain highlight the importance hu-CD34+ NSG mice model (refer to ¹⁸⁻²⁰). In the pioneer study done by Gorantla et. al. ²¹ on the development of this model p24 staining in various brain sections were shown along with markers of HIV-1 infected human macrophages. We have added Supplementary Fig 10 and 11 which demonstrates active HIV-1 infection in brain of our CD34+ NSG mice cohort, which is in agreement with other studies^{21,22} and demonstrates that this model can be used to evaluate anti-HIV modalities in the brain when delivered using an exosome-delivered ZPAMt.

Furthermore, short term oral cART treatment showed a controlled rise in viremia to ~ 2 to 3 log fold (Figure 5B). In the absence of no-ART control, we cannot comment to what extent cART worked or not. However, the experimental approach utilized here was to determine whether ZPAMt-exosomes suppress virus in adjunct to ART. As this was our experimental objective, based on our ethics and IACUC to only utilize mice required for the experimental objective and hypothesis testing, we did not include a no cART control. In essence to include the no cART mouse control would be in violation of the 3-R guideline of our IACUC at the City of Hope. Additionally, we agree that oral cART is variable (12 out of 25) show viremia increase up to ~1-to-2-fold change while the remaining mice showed ~3 to 4 fold change at week 4 vs week 2). This is the result of the variability in the mouse's consumption of the cART medicated water. As we observed in these series of experiments, mice recovering from anesthesia and healing from RO bleeding prefer diet-gel® for both their hydration and food instead of medicated water containing cART. For reviewer's consideration CD4+ cells rescue data in hu-CD34+ NSG mice has now been added as Supplementary Fig 12c. We see significant increase in CD4+ cells at week 10 compared to week 2 in all mice and a slight loss of CD4+ in nLuc exosomes treated mice compared to ZPAMt treated mice (Supplementary figure 12c). The decline in viremia due to ZPAMt packed exosome treatment increases in significance when we baseline it to absolute CD4+ cells counted from hu- CD34+ NSG mice at week 10 as suggested by the reviewers (Supplementary figure 12e). Though it is difficult to comment on cell specific uptake of exosome we are confident of CD4+ cells (Jurkat cells) and PBMCs collectively take up exosomes (refer to Supplementary Fig 4).

What are the levels of viral RNA in the spleen and gut tissue?

Response: We have added splenic virus levels in CD34+ mice cohort in Fig. 5e to the manuscript. Regrettably we did not save gut tissues for HIV load detection.

Reviewer #2 (Remarks to the Author):

The manuscript by Surya Shrivastava and colleagues titled, "Exosome-mediated stable epigenetic repression of HIV-1" describes an in-depth study whereas virus infected cells are targeted for suppression by epigenetic means to induce a stable "block and lock" inhibition for viral-gene expression. An HIV-1 promoter was developed to target a Zinc Finger Protein (ZFP-362) shown to bind to an NFκB doublet on the HIV-1 LTR. ZFP-362 was fused to a spectrum of domains of DNA methyltransferase 3A to improve viral suppression efficiency. Exosomes were engineered to package and deliver this therapeutic cargo. Results were affirmed in humanized mouse models of HIV-1 infection. The works serve to suggest that exosome-mediated systemic delivery platform for therapeutic cargos to epigenetically repress HIV-1 infection can provide a novel therapeutic directive to combat ongoing HIV-1 infection/replication.

While the results are of potential interest there are a number of major concerns for this work in its current form.

Response: Thank you for recognizing the impact of this work and taking the time to provide insights and considered comments and suggestions.

First, the specificity of ZPAMt to stably represses HIV-1 in a long term and specific manner by targeting CpG methylation of the 5' LTR warrants clarifications.

Response: The reviewer's curiosity for ZPAMt's mechanism of action is highly appreciated. In the preceding paper from our lab²³, the importance of the LTR 362 site on HIV-1 LTR promoter in context of gene transcription and expression was demonstrated. In a more recent paper from our lab, we describe the development and characterization of the ZFP-362b protein which can specifically bind at site 362 in HIV-1 LTR promoter¹. This LTR-362 site is comprised of NFκB binding site which is unique to LTR and thus adds to specificity of ZFP domain containing proteins. Details of this site and ZFP-362b binding are described in recent paper¹. In the current manuscript we confirmed that myc-tagged ZFP-362b and ZFP-362b fusion proteins like ZD3A, ZKMT and ZPAMt bind to the same site as determined by CHIP assay using anti-myc antibody for immunoprecipitation and designing specific primers towards LTR encompassing 362 site (Figure 2b). As per reviewer's suggestion we have amended the text to provide a brief description of ZFP's binding specificity. Further, we have provided extensive data characterizing the suppressive mechanism of the ZPAMt. We have shown that the ZPAMt is expressed (Fig. 1g), bind to the LTR (Figure 2b) and imparts CpG methylation at the site (Fig. 2e) and in vivo (Fig. 4d). This results in suppression of HIV in vitro (Fig. 2d) and in vivo (Fig 4b-e) and Fig (5b-e) and was able to suppress virus in vitro for up 60 days invitro (Fig 2g) Importantly as we included ZFP control we affirmed that repression is PAMt domain specific effect. Collectively this data demonstrates mechanistically the mode of suppression of HIV is targeted via the ZFP and implemented via PAMt domain.

The fact that the repression is relatively resistant to reactivation is shown in one plate of a single experiment with viral measures at the transcriptional level. These results should be affirmed over daily time periods with measures of progeny virus (reverse transcriptase activity) production, viral protein synthesis along with viral RNA measures.

Response: We appreciate the reviewer's experimental approach here in juxtaposing several approaches to measure HIV expression. We reasoned that measuring Gag RNA will best reflect whether ZPAMt is able to exert transcriptional control. For reviewer's consideration time course of ZD3A mediated repression has now been added in Supplementary Fig. 1c, measurement of viral p24 measurement and viral copy number have now all been added to this resubmission (Supplementary Figs 1a and b). Additionally, as per Supplementary Fig 1a and b we are confident that transcriptional repression corroborates with the observed reduction of viral copy number and viral protein as measured by p24 ELISA and thus is not redundantly repeated. Furthermore, all experiments were done in triplicate simultaneously and the results of this work is presented

These cross-validating experiments are important in that it forms the basis for why ZPAMt was selected and used to make it deliverable to virus infected cells through exosomes. Re-analysis of the virological tests should include additional biological replicates and positive controls beyond what is now illustrated for ZFP-362. These additional assays will affirm these critical data sets.

Response: We completely agree with the reviewer here regarding the impact of cross-validating experiments to look at HIV expression. We provided extensive data here Fig 1b-d, f-h, Fig 2a-h, Fig 3 f-g, Fig 4 b-d Fig 5 b-e and Supplementary Fig. 3, and 7 as well as recently published works on the ability of ZFP-362 to target and modulate HIV transcription¹ (now added to this resubmission), to be exceedingly convincing with regards to the antiviral impact of ZPAMt. The rationale for this is that we present not only antiviral expression analysis but also delineate ZPAMt directed CpG methylation and epigenetic silencing to the LTR. Notably this manuscript is the first demonstration of exosome delivery CpG methylation of HIV in the brain and bone marrow, a huge observation portending the utility of exosomes to target difficult to access organs. The juxtaposition of different experimental approaches to explain an observation is always preferable compared to multiple analysis of the same factor, as a greater depth of understanding and mechanistic insights can ensue. For reviewer's consideration substantial additional data to confirm ZPAMt's HIV-1 repressor activity has now added in Supplementary Fig. 1,2, and 12

Second, while the exosome system used to deliver mRNA for the epigenetic inhibitor of HIV-1 report that they can induce an epigenetic "block and lock" of HIV-1 expression the results given are not uniform in that different donor PBMC show variant results. These data sets should be affirmed with multiple viral strains given at different multiplicities over varied time points and beyond viral RNA for measures of actual virus production through RT activity measures in culture fluids. The data may also be replicated by using a transformed cell line in measurements.

Response: CHI Jurkat cells used in the study are stably infected with pNL4-3 derived HIV-1 and in the mouse studies we used pNL4-3 BaL derived HIV-1. As CHI Jurkat cells are already harboring HIV-1 we cannot change multiplicity of infection. Our recent paper published on characterization of ZFP demonstrates that ZFP fusion proteins have conversed effects across different viral strains and cell lines¹. Also, the use of two independent HIV mice models used here have human tissue derived from different deidentified donors, and therefore reinforces the utility of ZPAMt across multiple donors and donor variability. For the reviewer's consideration we have now added the repressive effects of ZPAMt on different cell lines as well in Supplementary Fig 2.

Third, while the observations purport to provide proof of principle demonstration for the utility of exosomes for epigenetics-based therapeutics in vivo the results seen in humanized mice and for both PBMC and CD34 reconstituted mice are highly variable. An explanation should be provided for the different results seen and the variant experimental protocols used.

Response: We appreciate reviewer's acknowledgement of present work as a proof of principle demonstration however, we disagree with reviewer's observation in variability of results between the two mouse models of HIV used. In spite of differences in human donor tissues, methodologies and progression of infection we consistently observe that exosomes with ZPAMt mRNA are able to significantly suppress viremia (Figs 4b and 5b). It's the inherent difference in these mouse models that we see active HIV-1 infection (and its repression) in the central nervous system of hu-CD34+ mice but not in hu-PBMC NSG mice²¹. Differences in the protocol is only pertaining to the methodology required to establish the PBMC or CD34+ cell transplant in mice, otherwise the ZPAMt testing methodology is same for both models. Those papers which have done the pioneer work in establishing these mouse models have been cited in manuscript.

A secondary concern is the transient use of antiretroviral therapy for one to two weeks without measured drug levels and evidence of complete viral suppression. This questions the results for the conclusions of stable targeted modulation of gene expression.

Response: Although we don't see ART mediated eradication of HIV, the use of cART for two weeks prevented exceedingly high levels of infection. In a preliminary study, now added as a supplementary data 'no cART' mice after two weeks of infection showed ~3 to 4 log fold higher viremia (Supplementary Fig 6). Although use of oral ART treatment suppressed virus (Supplementary Fig 6) short term oral cART treatment resulted in -1 to 1 log fold change (Figure 4b) or a rise in viremia to only about ~2 to 3 log fold (Figure 5b). For the reviewer's consideration an additional figure comparing median viremia progression in ART treated, noART and ART + control or therapeutic exosome is added (Fig 4c) which demonstrates that only ZPAMt exosome treatment imparts stable suppression of viremia in HIV-1 infected hu-PBMC NSG mice, even longer than cART treated mice. However, it should be noted by the reviewers and editor that the experimental approach utilized here was not to determine the efficacy of ART, which is

well established in the literature, but rather to determine whether ZPAMt-exosomes suppress virus in adjunct to ART.

Differences are observed in bone marrow and spleen which are not easily understood but need be explained.

Response: For the reviewers question about bone marrow and spleen virus levels, if we follow the median of fold change of Gag RNA in bone marrow and spleen cells from hu-PBMC NSG mice we can see strong reduction of HIV-1 transcripts due to ZPAMt packed exosome treatment on mice. Descriptions are now elaborated in manuscript as per the reviewer's suggestion.

Fourth, the data set illustrated for the brain is difficult to interpret as there is no reconstitution of human cells in the brain and the only cells seen in these animals are cells that migrate from the periphery to the CNS. No primary neural cells are of human origin. There again needs to be some explanation as to what is being recorded in the brain experiments.

Response: The reviewer has correctly pointed out that in the hu-CD34+ NSG mouse model infected macrophages infiltrate the CNS²¹. Currently, the HIV reservoir in CNS resident macrophages appears to be the biggest challenge for HIV-1 management as the virus remains hidden in ART inaccessible regions of the brain²⁴. As such, from the hu-CD34+ NSG mouse model, we have made sections of formalin fixed paraffine embedded (FFPE) brain tissue and stained for p24 to show signs of HIV-1 infection which have now been added for reviewer's consideration in Supplementary Fig 10 and 11. Also we isolated RNA from FFPE which was incredibly low in yield and assessed with Taqman probes to quantitate RNA from Pol region instead of LTR refer to Figure 5C.

Fifth, in order to achieve stable transcriptional repression of HIV-1 the authors developed a vector expressing recombinant fusion protein, ZFP 362b-DNMT3A (ZD3A) this was reported as being capable of binding to the NF-kB doublet binding site within LTR promoter of HIV-1 and fused to a DNA methyl transferase 3 domain (3A) to generate. Once again, a more complete description of these constructs and its development is in order.

Response: We apologize for this oversight. The molecular characterization of ZFP362 is now published¹. We have amended this resubmission to cite this manuscript and discuss the ZFP accordingly.

Sixth, I do not see how a transfection assay can be monitored for 10 days? What is the evidence for gene expression to continue this long? The transfection of ZD3A into HIV-1 infected Jurkat cells (CHI-Ju) is noted but the levels of viral production (for example, RT activity in culture supernatant fluids) was not shown. The reduced HIV-1 expression should be shown over time by more than one viral detection system.

Response: Jurkat cells being a suspension cell line were transfected and maintained in 10% FBS containing RPMI. Every two to three days, half of the cells were discarded and equal volume of fresh media was added. This way transfected cells were maintained not only for 10 days but for sixty days. Most definitely gene expression from plasmid is lost but the effect of ZD3A in exerting transcriptional repression takes a while to buildup and becomes significant at day 10 post-transfection (Figure 1D, F, H). As we were aiming to see transcription repression mediated by ZFP fusion proteins, gag RNA qRT-PCR was the chosen readout. For the reviewer's consideration we have now added viral copy number/cell and p24 readouts in (Supplementary Fig 1) for key experiments which notably corroborates well with the gag RNA readouts provided in manuscripts earlier submission. Repressive effects of ZPAMt on different cell lines was also observed and is now reported (Supplementary Fig. 2)

The TNF induction was confusing as why was TNF needed if the cells are continuously releasing virus?

Response: As for the reviewer's remark on TNF α addition, it is known to decondense the LTR promoter for transcription factors to access²⁵ or in our case it seems to facilitate access to ZFP fusion proteins. We can see sooner onset of LTR repression via ZD3A in TNF α pretreated cells compared to non-treated cells (Figures 1D,E).

Was another cell line used? If so how was it characterized? Was PMA, LPS or other inducers used to affirm the test results? The TNF appears to be a single time point and as such was this affirmed over time and if so what time points were measured? Why was a model of HIV-1 infection of microglial cells used as in none of the humanized mice are microglial cells of human origin present?

Response: To validate ZPAMt mediated repression across cells of non-hematopoietic origin, microglial cells characterized in²⁶ were also tested in the initial stage of project (Figure 1H). For the reviewer's consideration we have now added (Supplementary Fig 2) which shows the effect of ZPAMt on different latent cell lines. PMA and LPS and other inducers were not used, as they did not provide any additional insights to what we were asking in the current context. In the current scope of work however we selected those inducers that modulate the transcriptional and epigenetic states, e.g. one working on methylation level, one working on HDAC inhibition and one over all NF- κ B inducer and thus Azacytidine, TSA and TNF α were used. Like TNF α , PMA and LPS also act via the NF κ B site^{27,28}. As for the reviewer's concern about TNF α , it is well-known to show activation within 12 hours, so no other time points were used. The reviewer has correctly pointed out that microglial cells in mouse are not of human origin. HIV-1 infection of microglial cells in Fig 1H is only test ZPAMt's efficacy in non-hematopoietic, HIV infectible cell line.

Reviewer #3 (Remarks to the Author):

The manuscript by Shrivastava et al., provided an interesting approach to suppress the

replication of integrated HIV provirus. Epigenetic modulation with selected fusion protein delivered by exosomes also is an attractive approach.

However, it is not as practical as of today. Delivery of foreign proteins will induce immune responses and minimize therapeutic efficacy.

Response: The reviewer is correct that this represent a proof-of-concept study but represent a significant development of a novel, targeted “block-and-lock” approach with a next-generation exosome delivery system, and describes a noteworthy and unique addition to treatment strategies for HIV. However, this is exactly why this research should be considered for publication in Nature communications, as the innovative work and the data presented here provide proof of principle evidence that exosomes-based therapeutics are feasibly and portend a future application to not only treat HIV but an application that can be used for several other human diseases. Notably, several fusion proteins and exosome-based therapeutics are already under clinical trial for a variety of diseases including Covid-19 recently²⁹⁻³².

There are some technical problems with in vivo studies on humanized mice. The suppression of HIV replication by multiple injections of exosomes with selected construct is not convincing.

Response: We observed significant suppression of viremia in mice when we measured on week 10 after infection as depicted in Figure 4B and Figure 5B. We also observed significant decline in viral Gag RNA levels in brain, bone marrow and spleen in CD34+ mouse cohorts as depicted Figures 5c, d and e. Additionally ZPAMt exosomes treated PBMC mice cohort show a strong trend in decline in viral Gag RNA levels bone marrow and spleen tissue, add in Supplementary Fig.4 (a) and (b).

There are some problems with the results. Figure 4. One week of ART treatment not clear what for and not explained in the text.

Response: Reviewer 3 has rightly pointed our oversight about the lack of a suitable explanation for using two weeks of oral cART. Administration of Oral ART in HIV-1 infected hu-PBMC-NSG mice was shown to prevent loss of CD4+ cells (recently added Supplementary Fig 6b). We hypothesized that two weeks of oral ART administration will decelerate loss of CD4+ cells and subsequent administration of exosomes will suppress viremia. In absence of two weeks of oral cART, due to the rapid loss of CD4 T-cells, not enough CD4+ cells will be left in murine system to evaluate effects of the anti-HIV exosomes. With extended administration of oral cART, HIV-1 will more closely resemble treatments and viral infections in humans and the virus might go under indefinite latency. We have now added these data substantiating this notion in Supplementary Fig. 6 and Fig. 4c. In light of these observations and since we wanted to keep the experimental parameters the same for the two mouse cohorts we used the same methodology of administering two weeks of oral ART for hu-CD34+NSG mice as well. Explanation about short term ART usage has now been elaborated in manuscript.

The viral load declined spontaneously in control exosomes treated mice.

Response: Even though we see decline in control mice as (possible after effect of oral cART) the decline in viremia in ZPAMt treated mice were two log fold greater than control or ZFP treated mice as shown in Figure 4B and Figure 5B. Collectively, these data demonstrate that our approach works to repress HIV expression in vivo.

Induction of latency (no production of viral particles and new infection of available naïve CD4+ cells) in vivo in the presence of actively replicating virus is very questionable.

Response: The reviewer has keenly observed and is perhaps pointing to the discordance between tissue specific virus levels in bone marrow of hu-PBMC-NSG mice cohort and the hypermethylation of LTR promoter from viral DNA isolated from same bone marrow sample. Unfortunately, the discordance between LTR methylation and viremia levels is a natural aspect of HIV, as such discordance has been documented on short term ART treated people living with HIV³³. The data present here demonstrates a way to achieve exosome mediated methylation at specific LTR promoter, systemically. Perhaps in a few outliers of treated population like any therapy, it will require more than hypermethylation to induce true latency, although we see significant suppression of viremia and hypermethylation in the studies presented here.

Sometimes in hu-PBL mice with time virus destroy available CD4+ cells and viral load drop as no more targets for active replication and viral production.

Response: We understand the reviewer's thoughts here. However, if the above scenario were playing out in vivo then we would expect to see comparable viremia reduction in reporter and ZFP controls as well as those treated with ZPAMt. We observed a two to three log fold decline in viremia in ZPAMt treated mice as shown in Figures 4B and 5B. We have also included new data in Supplementary Fig. 12 which depicts the persistence of CD4+ cells until week 10 i.e. endpoint time of the experiment. Baselineing the viremia by absolute count of CD4+ cells in mice adds strength to our observation that repression in viremia is due to specific effect of ZPAMt and not loss of CD4+ cells.

Information about the absolute number of human CD4+ cells is important and needs to be included.

Response: As our primary aim was to report exosome mediated delivery of specific suppressor protein which can impart methyl tags on the HIV-1 LTR promoter in mouse models of HIV infection, we did not provide CD4+ data initially. However, we have now added these data in Supplementary Fig. 12 which depicts the persistence of CD4+ cells until week 10, the endpoint time of the in vivo experiments.

To confirm the reduction of HIV replication all HIVgag PCR values should be adjusted to the number of human cells in both models.

Response: As per reviewer's suggestion we have now added Supplementary Fig. 12 d and e to the manuscript.

Figure 5. The ART treatment did not result in suppression of HIV replication. The strange result with an increased viral load from 2 to 4 weeks post-infection. In original reference #52, which was used as a protocol for the treatment of mice "Viremic hu-NSG mice exhibited a robust suppression of viral load in response to the oral three-drug cART regimen within 2 weeks of treatment" with significant recovery of CD4+ cells.

Response: Although we don't see ART mediated eradication of HIV, the use of ART prevented exceedingly high levels of virus and more closely represents ongoing HIV infection in humans on ART. 'No cART' treated mice after two weeks of infection showed ~3 to 4 log fold higher viremia Supplementary Fig. 6a. While one week of oral ART treatment completely suppressed virus in Supplementary Fig. 6a, short term oral cART treatment showed a rise in viremia to about ~2 to 3 log fold Figure 5b. We cannot comment on the effects of viral infection in a no-ART treated mouse as to what extent ART worked or not. The experimental approach utilized here was to determine whether ZPAMt-exosomes suppress virus in adjunct to ART. As this was our experimental objective, based on our ethics and IACUC to only utilize mice required for the experimental objective and hypothesis testing, we did not include a no ART control for CD34+ mice cohort. In essence to include the no ART mouse control would be in violation of the 3-R guideline of our IACUC at the City of Hope. Additionally, we agree that oral cART is variable (12 out of 25 mice show viremia change up to -1 to 2 fold change and rest show 3 to 4 fold change at week 4 vs week 2) as it is subjected to variability in mouse's consumption of medicated water. As we observed in these series of experiments the mice, following recover from anesthesia and healing from RO bleeding, tended to prefer the diet-gel® for both hydration and food instead of medicated water containing cART. We saw that CD4+ cell population was sustained in CD34+ mice cohort till the last time point week 10 of viremia progression (Supplementary data 12c). For reviewer's consideration we have also presented viremia baselined with absolute CD4+ count, which shows highly significant decline in viremia in CD34+ mice due to ZPAMt-packed exosomes treatment.

Material and Methods need to provide information about the infection of humanized NSG mice. Reference #52 has at least three HIV strains. Which one and at what dose did the authors use?

Response: Reference#52 has only used one pNL4-3 BaL derived HIV-1 whose NIH AIDS repository catalogue number ARP-11440 is added to the manuscript. As already described in manuscript 200 ng equivalent of p24 was used for 10⁷ activated PBMCs and was injected intraperitoneally in each mouse for obtaining hu-PBMC NSG mice. For CD34+ mice cohort 200 ng equivalent of p24 was injected into each mice.

Minor mistake:

Reference 19 is not Kojima et al. It is #27.

Response: We have amended this to correctly reflect the proper citation.

Literature cited

- 1 Scott, T. S., D. O'meally, N.A. Grepo, M.S. Weinberg, V. Planelles, and K.V. Morris. Broadly active zinc finger protein-guided transcriptional activation of HIV-1. *Molecular Therapy Methods & Clinical Development*, doi:<https://doi.org/10.1016/j.omtm.2020.10.018> (2020).
- 2 Berro, R. *et al.* Acetylated Tat regulates human immunodeficiency virus type 1 splicing through its interaction with the splicing regulator p32. *J Virol* **80**, 3189-3204, doi:10.1128/JVI.80.7.3189-3204.2006 (2006).
- 3 Thery, C. *et al.* Minimal information for studies of extracellular vesicles 2018 (MISEV2018): a position statement of the International Society for Extracellular Vesicles and update of the MISEV2014 guidelines. *J Extracell Vesicles* **7**, 1535750, doi:10.1080/20013078.2018.1535750 (2018).
- 4 Smith, K. M., Himiari, Z., Tsimbalyuk, S. & Forwood, J. K. Structural Basis for Importin-alpha Binding of the Human Immunodeficiency Virus Tat. *Sci Rep* **7**, 1650, doi:10.1038/s41598-017-01853-7 (2017).
- 5 Lange, A. *et al.* Classical nuclear localization signals: definition, function, and interaction with importin alpha. *J Biol Chem* **282**, 5101-5105, doi:10.1074/jbc.R600026200 (2007).
- 6 Saunderson, S. C., Dunn, A. C., Crocker, P. R. & McLellan, A. D. CD169 mediates the capture of exosomes in spleen and lymph node. *Blood* **123**, 208-216, doi:10.1182/blood-2013-03-489732 (2014).
- 7 Cui, X. *et al.* Circulating Exosomes Activate Dendritic Cells and Induce Unbalanced CD4+ T Cell Differentiation in Hashimoto Thyroiditis. *J Clin Endocrinol Metab* **104**, 4607-4618, doi:10.1210/jc.2019-00273 (2019).
- 8 Danesh, A. *et al.* Exosomes from red blood cell units bind to monocytes and induce proinflammatory cytokines, boosting T-cell responses in vitro. *Blood* **123**, 687-696, doi:10.1182/blood-2013-10-530469 (2014).
- 9 Fu, W. *et al.* The Secreted Form of Transmembrane Protein 98 Promotes the Differentiation of T Helper 1 Cells. *J Interferon Cytokine Res* **35**, 720-733, doi:10.1089/jir.2014.0110 (2015).
- 10 Hong, X., Schouest, B. & Xu, H. Effects of exosome on the activation of CD4+ T cells in rhesus macaques: a potential application for HIV latency reactivation. *Sci Rep* **7**, 15611, doi:10.1038/s41598-017-15961-x (2017).
- 11 Hu, W., Song, X., Yu, H., Sun, J. & Zhao, Y. Released Exosomes Contribute to the Immune Modulation of Cord Blood-Derived Stem Cells. *Front Immunol* **11**, 165, doi:10.3389/fimmu.2020.00165 (2020).
- 12 Liu, H. *et al.* Exosomes derived from dendritic cells improve cardiac function via activation of CD4(+) T lymphocytes after myocardial infarction. *J Mol Cell Cardiol* **91**, 123-133, doi:10.1016/j.yjmcc.2015.12.028 (2016).
- 13 Matsumoto, K. *et al.* Exosomes secreted from monocyte-derived dendritic cells support in vitro naive CD4+ T cell survival through NF-(kappa)B activation. *Cell Immunol* **231**, 20-29, doi:10.1016/j.cellimm.2004.11.002 (2004).

- 14 They, C. *et al.* Indirect activation of naive CD4⁺ T cells by dendritic cell-derived exosomes. *Nat Immunol* **3**, 1156-1162, doi:10.1038/ni854 (2002).
- 15 Xu, Y. *et al.* Macrophages transfer antigens to dendritic cells by releasing exosomes containing dead-cell-associated antigens partially through a ceramide-dependent pathway to enhance CD4⁽⁺⁾ T-cell responses. *Immunology* **149**, 157-171, doi:10.1111/imm.12630 (2016).
- 16 Zhang, B. *et al.* Mesenchymal stromal cell exosome-enhanced regulatory T-cell production through an antigen-presenting cell-mediated pathway. *Cytotherapy* **20**, 687-696, doi:10.1016/j.jcyt.2018.02.372 (2018).
- 17 Zhang, L. *et al.* Plasma Transfusion Promoted Reprogramming CD4⁽⁺⁾ T Lymphocytes Immune Response in Severe Sepsis Mice Model Through Modulating the Exosome Protein Galectin 9. *Cell Transplant* **29**, 963689720947347, doi:10.1177/0963689720947347 (2020).
- 18 Evering, T. H. & Tsuji, M. Human Immune System Mice for the Study of Human Immunodeficiency Virus-Type 1 Infection of the Central Nervous System. *Front Immunol* **9**, 649, doi:10.3389/fimmu.2018.00649 (2018).
- 19 Honeycutt, J. B. & Garcia, J. V. Humanized mice: models for evaluating NeuroHIV and cure strategies. *J Neurovirol* **24**, 185-191, doi:10.1007/s13365-017-0567-3 (2018).
- 20 Mallard, J. & Williams, K. C. Animal models of HIV-associated disease of the central nervous system. *Handb Clin Neurol* **152**, 41-53, doi:10.1016/B978-0-444-63849-6.00004-9 (2018).
- 21 Gorantla, S. *et al.* Links between progressive HIV-1 infection of humanized mice and viral neuropathogenesis. *Am J Pathol* **177**, 2938-2949, doi:10.2353/ajpath.2010.100536 (2010).
- 22 Satheesan, S. *et al.* HIV Replication and Latency in a Humanized NSG Mouse Model during Suppressive Oral Combinational Antiretroviral Therapy. *J Virol* **92**, doi:10.1128/JVI.02118-17 (2018).
- 23 Saayman, S. M. *et al.* Potent and Targeted Activation of Latent HIV-1 Using the CRISPR/dCas9 Activator Complex. *Mol Ther* **24**, 488-498, doi:10.1038/mt.2015.202 (2016).
- 24 Kolson, D. Neurologic Complications in Persons With HIV Infection in the Era of Antiretroviral Therapy. *Top Antivir Med* **25**, 97-101 (2017).
- 25 Pearson, R. *et al.* Epigenetic silencing of human immunodeficiency virus (HIV) transcription by formation of restrictive chromatin structures at the viral long terminal repeat drives the progressive entry of HIV into latency. *J Virol* **82**, 12291-12303, doi:10.1128/JVI.01383-08 (2008).
- 26 Alvarez-Carbonell, D. *et al.* Toll-like receptor 3 activation selectively reverses HIV latency in microglial cells. *Retrovirology* **14**, 9, doi:10.1186/s12977-017-0335-8 (2017).
- 27 Osborn, L., Kunkel, S. & Nabel, G. J. Tumor necrosis factor alpha and interleukin 1 stimulate the human immunodeficiency virus enhancer by activation of the nuclear factor kappa B. *Proc Natl Acad Sci U S A* **86**, 2336-2340, doi:10.1073/pnas.86.7.2336 (1989).

- 28 Equils, O. *et al.* Bacterial lipopolysaccharide activates HIV long terminal repeat through Toll-like receptor 4. *J Immunol* **166**, 2342-2347, doi:10.4049/jimmunol.166.4.2342 (2001).
- 29 Dhankhar, R., Gupta, V., Kumar, S., Kapoor, R. K. & Gulati, P. Microbial enzymes for deprivation of amino acid metabolism in malignant cells: biological strategy for cancer treatment. *Appl Microbiol Biotechnol* **104**, 2857-2869, doi:10.1007/s00253-020-10432-2 (2020).
- 30 Cangini, D. *et al.* Tagraxofusp and anti-CD123 in blastic plasmacytoid dendritic cell neoplasm: a new hope. *Minerva Med* **111**, 467-477, doi:10.23736/S0026-4806.20.07018-4 (2020).
- 31 Dai, S. *et al.* Phase I clinical trial of autologous ascites-derived exosomes combined with GM-CSF for colorectal cancer. *Mol Ther* **16**, 782-790, doi:10.1038/mt.2008.1 (2008).
- 32 Reza khani, L., Kelishadrokh, A. F., Soleimanizadeh, A. & Rahmati, S. Mesenchymal stem cell (MSC)-derived exosomes as a cell-free therapy for patients Infected with COVID-19: Real opportunities and range of promises. *Chem Phys Lipids* **234**, 105009, doi:10.1016/j.chemphyslip.2020.105009 (2021).
- 33 Cortes-Rubio, C. N. *et al.* Longitudinal variation in human immunodeficiency virus long terminal repeat methylation in individuals on suppressive antiretroviral therapy. *Clin Epigenetics* **11**, 134, doi:10.1186/s13148-019-0735-9 (2019).

REVIEWER COMMENTS

Reviewer #2 (Remarks to the Author):

The authors addressed my comments in a satisfactory manner.

Reviewer #3 (Remarks to the Author):

I am not convinced with the claim of HIV-1 suppression in the brain.

In CD34-humanized mice authors had very low levels of HIV-1 infection 10^3 - 10^4 at starting point. The two log increase during ART therapy means that ART was not efficient at all.

Figure 5b. The brain infection and measurement of pol RNA also is confusing. If mouse have in peripheral blood 10^3 copies/ml and reduction 100 or 1000 times means that authors were able detect 1 pol copy by RT-PCR Fig 5c? Without perfusion it is difficult to dissect blood and tissue viral RNA content.

If CD34-humanized mice have very low peripheral viral load, brain viral load will be also very low from the beginning in selected model.

It is important for authors to claim brain accessibility of exosomes and efficacy of viral replication repression, however provided evidences are not convincing due to technical parameters of the model.

To prove ability of exosome to enter brain and repress human microglia HIV-1 infection authors need different model.

The real repression will be undetectable levels of HIV RNA in blood and tissues. Authors did not provide information about detection limit of in house RT-PCR for HIV-1 gag RNA.

The supplemental figures 10 and 11 are not convincing. Brain sections in Figure 10 showed positive for p24 staining of neurons, which need explanation.

Brown inclusions in Figure 11 also are not HIV-1p24 protein.

“**Response to reviewers**” comments: We appreciate the reviewers time reviewing this manuscript. Our response is below in *italics* and underlined for clarity.

Reviewer #3 (Remarks to the Author):

I am not convinced with the claim of HIV-1 suppression in the brain. In CD34-humanized mice authors had very low levels of HIV-1 infection 10^3 - 10^4 at starting point. The two log increase during ART therapy means that ART was not efficient at all.

Response: While it may seem like a 2 week course of ART was not sufficient to reduce viral load significantly, this is common in the oral ART mouse model as some mice will drink less medicated water some days/weeks versus others. In the present study, where the aim was to study exosome mediated repression of HIV, we did not need the suppressed viral load to test efficacy of the ART treatment, rather we needed productive viremia and infection in the cells and ART allows for repression of the HIV burst phase and therefore the viral infection to occur at the time of Exosome treatment. Additionally, as shown in fig 4C without ART, viremia is found to increase to 10^5 - 10^6 copies/ml and remain that high, leading ultimately to exhaustion of CD4+ T cells which would not be ideal for testing the ability of therapeutic to control HIV expression, and substantiates our rationale for the 2 week ART treatment. So, in conclusion the data presented here suggests that 2 weeks of ART allowed for a suitable reduction in the HIV burst phase for the successfully testing of our hypothesis that ZPAMt repressor loaded exosomes can repress HIV in vivo.

Figure 5b. The brain infection and measurement of pol RNA also is confusing. If mouse have in peripheral blood 10^3 copies/ml and reduction 100 or 1000 times means that authors were able detect 1 pol copy by RT-PCR Fig 5c? Without perfusion it is difficult to dissect blood and tissue viral RNA content.

Response: One full length cellular HIV-1 genomic RNA can yield 109 different splice variants² and thus 1 to 2 log fold change in ratio of full-length read through transcript to Pol specific transcript was observed from infected cells³. In the present study the brain infection was measured after HIV infection progressed for 10 weeks in mice. Peripheral blood copy numbers cannot be equated with formalin fixed paraffin embedded brain tissue (FFPE) sections. As explained in the methods section, to calculate extracellular viremia copy number from serum, primers specific for LTR region were used to detect the genomic transcripts in the serum. For detecting the virus in brain, RNA was extracted from FFPE tissue section which also contained cellular RNA (both mouse and potentially also human). Pol specific primers detected shorter viral transcripts present in the infected cell which is expected to be much higher in concentration compared to genomic HIV-1 full-length LTR containing transcripts. As such Pol specific primers are much more sensitive in detecting HIV-1 transcripts but are incapable of confidently

quantitating viremia, rather they show you the presence and copies of HIV Pol transcripts. As our aim was to detect transcriptional repression and not quantitating total HIV-1 copies from brain tissue we have used these Pol specific primer in Fig 5C. Notably, the suitable controls were also used.

It is widely known in literature that HIV-1 found in the brain is not transmitted to the brain through the blood viremia but rather appears to be transmitted to the brain through the infiltration of infected monocytes which become resident macrophage/microglial cells in brain and act as reservoir of HIV-1⁴⁻⁶. Due to existence of definite blood brain barrier in mice models of HIV-1 infection, p24+ staining of blood borne free HIV in brain tissue can be ruled out. Although perfusion was not performed, mice were bled to collect 700µl of blood via cardiac puncture after anesthesia as per routine euthanasia protocol. Thus, we do not suspect blood viremia are providing the p24+ staining in FFPE brain tissue or contributing to RNA extracted from brain FFPE tissues. As the CD34+ mouse model is widely accepted model for studying brain HIV-1 infection, we did not perform in-depth experiments to re-establish this model, rather we followed the work of others and were able to recapitulate the model as it's been described in the literature and were able to use this model to establish that exosomes act as an efficient carrier for our novel anti-HIV therapeutic moiety and repress HIV. Notably, we also used the SCID-PBMC mouse model and report similar findings. It is rare to see two HIV mouse models used in one manuscript, but we chose to use both models as the exosome mediated repression or HIV has never been reported before and two models offer more insights into the parameters of our approach.

If CD34-humanized mice have very low peripheral viral load, brain viral load will be also very low from the beginning in selected model. It is important for authors to claim brain accessibility of exosomes and efficacy of viral replication repression, however provided evidences are not convincing due to technical parameters of the model. To prove ability of exosome to enter brain and repress human microglia HIV-1 infection authors need different model.

Response: The potential of exosome to penetrate blood brain barrier has also been demonstrated in several other studies⁷⁻⁹. Exosome mediated delivery of reporter and ZFPAM cargo is shown, relative to negative controls, in those data presented in Supplementary Fig 5 and 9 respectively. The paper describing CD34+NSG HIV-1 mouse model¹⁰ is cited over 50 times including 19 papers wherein their methodology is adopted by researchers or more than 20 reviews where it is regarded as an apt for studying apt model of HIV neuropathogenesis. As the CD34+NSG mouse model is widely acclaimed to study HIV-1 in brain¹¹⁻¹⁵, the effect of ZPAMT packed exosomes were tested in this and the huPBMC+NSG mouse models. There are other models to study HAND, HAD for macaques, however these are cost prohibitive models to use in proof of concept studies and primates should always be used sparingly and in cases when other models cannot be used, this is not such a case. We would argue that the mouse models used here are well suited for those studies developed here and to test our hypothesis. While no animal model is perfect, the

use of 2 distinct HIV mouse models, rendering similar observations, suggests that exosomes functionally delivery anti-HIV modalities to transcriptionally repress HIV expression in vivo.

The real repression will be undetectable levels of HIV RNA in blood and tissues. Authors did not provide information about detection limit of in-house RT-PCR for HIV-1 gag RNA.

Response: Limit of detection (LOD) is an arbitrary value which varies as per technology and reagent used so we relied on statistical significance of results more than LOD. For the reviewer's consideration we have added the LOD in Fig 4b-c and Fig 5b where absolute viremia of mice are shown. Although, after scouring the literature we yet to come across and literature that is concordant with reviewer's comment that real repression will be undetectable levels of HIV RNA in blood and tissues. Reduction in viremia as well as tissue specific HIV-1 transcripts in Figure 4B, 5B-E is statistically significant. These reductions can be attributed to repression of LTR driven transcription due to LTR directed DNA methylation Figure 4D. As per reviewer's suggestion limit of detection is added in Figure 4B and Figure 5B where absolute viremia is measured and described in methods section.

The supplemental figures 10 and 11 are not convincing.

Response: Regrettably these are the data, e.g. the data are the data. While we cannot change the staining data, we hope that the inclusion of the positive control for the p24 antibody, as well as the inclusion of the uninfected NSG mouse control indicates that the staining is most likely representative of p24+ staining. We have also updated supplemental Figure 11 to include the NSG mouse control to show the specificity of this antibody.

Brain sections in Figure 10 showed positive for p24 staining of neurons, which need explanation.

Response: Thank you for this comment. Yes, while initially surprising these data may not be indicative of HIV-1 infection per say, but rather the accumulation of shed gp-120. It has been well established in the literature that HIV-1 proteins like gp120 are neurotoxic and are able to enter neurons through CXCR4 or through a lipid-raft mediated internalization process¹⁶⁻¹⁸. We postulate that we are observing shed gp-120 accumulation in the Purkinje cell layer that may be indicative of neuronal dysfunction. However, we do not test this assumption here and rather we only note that gp120 positivity is present in these sections, indicating that HIV gp120 is present in the mouse brains of this CD34 mouse model, which has been observed by others as well^{10,19}. Additionally, at this magnification and in absence of cell-type specific markers, one cannot make any claims that only the neurons are p24 positive, as there are no neuronal markers used here and any such suggestion is circumstantial. Our purpose of IHC was to learn whether after 10 weeks of HIV- infection progression to what extent we observe p24+ cells in brain FFPE tissue sections, which is exactly what we show in supplementary fig. 10 and 11.

Brown inclusions in Figure 11 also are not HIV-1p24 protein.

Response: As described above we have included an image of the negative control (an NSG mouse stained with p24 antibody). While we observe some diffuse background staining, it is not to the same level of staining observed in the HIV-1 infected mice in the same region. Furthermore, these data are supported by both Gorantla et al.,¹⁰ and Li et al.¹⁹ who observed p24 detection in the meninges of HIV-1 infected huCD34+ mice. As such we are confident that the “brown inclusions” are most likely indicative of a positive p24 signal.

Literature cited

- 1 Fosse, J. H., Haraldsen, G., Falk, K. & Edelmann, R. Endothelial Cells in Emerging Viral Infections. *Front Cardiovasc Med* **8**, 619690, doi:10.3389/fcvm.2021.619690 (2021).
- 2 Ocwieja, K. E. et al. Dynamic regulation of HIV-1 mRNA populations analyzed by single-molecule enrichment and long-read sequencing. *Nucleic Acids Res* **40**, 10345-10355, doi:10.1093/nar/gks753 (2012).
- 3 Moron-Lopez, S. et al. Human splice factors contribute to latent HIV infection in primary cell models and blood CD4+ T cells from ART-treated individuals. *PLoS Pathog* **16**, e1009060, doi:10.1371/journal.ppat.1009060 (2020).
- 4 Fischer-Smith, T. et al. CNS invasion by CD14+/CD16+ peripheral blood-derived monocytes in HIV dementia: perivascular accumulation and reservoir of HIV infection. *J Neurovirol* **7**, 528-541, doi:10.1080/135502801753248114 (2001).
- 5 Lamers, S. L. et al. The meningeal lymphatic system: a route for HIV brain migration? *J Neurovirol* **22**, 275-281, doi:10.1007/s13365-015-0399-y (2016).
- 6 Osborne, O., Peyravian, N., Nair, M., Daunert, S. & Toborek, M. The Paradox of HIV Blood-Brain Barrier Penetration and Antiretroviral Drug Delivery Deficiencies. *Trends Neurosci* **43**, 695-708, doi:10.1016/j.tins.2020.06.007 (2020).
- 7 Saeedi, S., Israel, S., Nagy, C. & Turecki, G. The emerging role of exosomes in mental disorders. *Transl Psychiatry* **9**, 122, doi:10.1038/s41398-019-0459-9 (2019).
- 8 Mizrak, A. et al. Genetically engineered microvesicles carrying suicide mRNA/protein inhibit schwannoma tumor growth. *Mol Ther* **21**, 101-108, doi:10.1038/mt.2012.161 (2013).
- 9 Kojima, R. et al. Designer exosomes produced by implanted cells intracerebrally deliver therapeutic cargo for Parkinson's disease treatment. *Nat Commun* **9**, 1305, doi:10.1038/s41467-018-03733-8 (2018).
- 10 Gorantla, S. et al. Links between progressive HIV-1 infection of humanized mice and viral neuropathogenesis. *Am J Pathol* **177**, 2938-2949, doi:10.2353/ajpath.2010.100536 (2010).
- 11 Mahmud, F. J. et al. Osteopontin/secreted phosphoprotein-1 behaves as a molecular brake regulating the neuroinflammatory response to chronic viral infection. *J Neuroinflammation* **17**, 273, doi:10.1186/s12974-020-01949-4 (2020).
- 12 Cornwell, W. D. et al. Tobacco smoke and morphine alter peripheral and CNS inflammation following HIV infection in a humanized mouse model. *Sci Rep* **10**, 13977, doi:10.1038/s41598-020-70374-7 (2020).
- 13 Dash, P. K. et al. HIV-1-Associated Left Ventricular Cardiac Dysfunction in Humanized Mice. *Sci Rep* **10**, 9746, doi:10.1038/s41598-020-65943-9 (2020).
- 14 Dash, P. K. et al. Sequential LASER ART and CRISPR Treatments Eliminate HIV-1 in a Subset of Infected Humanized Mice. *Nat Commun* **10**, 2753, doi:10.1038/s41467-019-10366-y (2019).

- 15 Honeycutt, J. B. & Garcia, J. V. Humanized mice: models for evaluating NeuroHIV and cure strategies. *J Neurovirol* **24**, 185-191, doi:10.1007/s13365-017-0567-3 (2018).
- 16 Avdoshina, V. *et al.* The HIV Protein gp120 Alters Mitochondrial Dynamics in Neurons. *Neurotox Res* **29**, 583-593, doi:10.1007/s12640-016-9608-6 (2016).
- 17 Cotto, B., Natarajanseenivasan, K. & Langford, D. HIV-1 infection alters energy metabolism in the brain: Contributions to HIV-associated neurocognitive disorders. *Prog Neurobiol* **181**, 101616, doi:10.1016/j.pneurobio.2019.101616 (2019).
- 18 Berth, S., Caicedo, H. H., Sarma, T., Morfini, G. & Brady, S. T. Internalization and axonal transport of the HIV glycoprotein gp120. *ASN Neuro* **7**, doi:10.1177/1759091414568186 (2015).
- 19 Li, W., Gorantla, S., Gendelman, H. E. & Poluektova, L. Y. Systemic HIV-1 infection produces a unique glial footprint in humanized mouse brains. *Dis Model Mech* **10**, 1489-1502, doi:10.1242/dmm.031773 (2017).
- 20 Chang, W. *et al.* Functionally distinct Purkinje cell types show temporal precision in encoding locomotion. **117**, 17330-17337, doi:10.1073/pnas.2005633117 %J Proceedings of the National Academy of Sciences (2020).
- 21 Avdoshina, V. *et al.* The HIV protein gp120 alters mitochondrial dynamics in neurons. *Neurotoxicity Research* **29**, 583-593 (2016).
- 22 Cotto, B., Natarajanseenivasan, K. & Langford, D. HIV-1 infection alters energy metabolism in the brain: Contributions to HIV-associated neurocognitive disorders. *Progress in Neurobiology* **181**, 101616, doi:<https://doi.org/10.1016/j.pneurobio.2019.101616> (2019).
- 23 Shah, A. & Kumar, A. HIV-1 gp120-Mediated Mitochondrial Dysfunction and HIV-Associated Neurological Disorders. *Neurotoxicity Research* **30**, 135-137, doi:10.1007/s12640-016-9619-3 (2016).
- 24 Kanmogne, G. D., Kennedy, R. C. & Grammas, P. HIV-1 gp120 Proteins and gp160 Peptides Are Toxic to Brain Endothelial Cells and Neurons: Possible Pathway for HIV Entry into the Brain and HIV-Associated Dementia. *Journal of Neuropathology & Experimental Neurology* **61**, 992-1000, doi:10.1093/jnen/61.11.992 %J Journal of Neuropathology & Experimental Neurology (2002).
- 25 Berth, S., Caicedo, H. H., Sarma, T., Morfini, G. & Brady, S. T. Internalization and axonal transport of the HIV glycoprotein gp120. *ASN Neuro* **7**, 1759091414568186, doi:10.1177/1759091414568186 (2015).
- 26 Teodorof-Diedrich, C. & Spector, S. A. Human Immunodeficiency Virus Type 1 gp120 and Tat Induce Mitochondrial Fragmentation and Incomplete Mitophagy in Human Neurons. **92**, e00993-00918, doi:10.1128/JVI.00993-18 %J Journal of Virology (2018).
- 27 Fields, J. A. *et al.* HIV alters neuronal mitochondrial fission/fusion in the brain during HIV-associated neurocognitive disorders. *Neurobiology of Disease* **86**, 154-169, doi:<https://doi.org/10.1016/j.nbd.2015.11.015> (2016).
- 28 Wächter, C., Eiden, L. E., Naumann, N., Depboylu, C. & Weihe, E. Loss of cerebellar neurons in the progression of lentiviral disease: effects of CNS-permeant antiretroviral therapy. *Journal of Neuroinflammation* **13**, 272, doi:10.1186/s12974-016-0726-0 (2016).
- 29 Kesby, J. P. *et al.* Cognitive deficits associated with combined HIV gp120 expression and chronic methamphetamine exposure in mice. *European Neuropsychopharmacology* **25**, 141-150, doi:<https://doi.org/10.1016/j.euroneuro.2014.07.014> (2015).
- 30 Rozzi, S. J. *et al.* Human Immunodeficiency Virus Promotes Mitochondrial Toxicity. *Neurotoxicity Research* **32**, 723-733, doi:10.1007/s12640-017-9776-z (2017).

REVIEWERS' COMMENTS

Reviewer #3 (Remarks to the Author):

The argument "data is data" does not prove that data are correct.

The brown depositions detected do not reflect HIV-1p24 presence. Such an artificial signal can happen on paraffin-embedded tissue sections but does not reflect the true presence of protein.

Reference 36 is a review article and does not provide experimental pieces of evidence of gp120 accumulation in neurons.

Reference 37 discuss mechanisms of gp120 internalization in cell culture.

Both references are not correct for positive staining for HIV-1p24 protein.

I cannot accept provided explanations and would like to recommend removing supplemental figures 10 and 11.

If authors claim that immune cells are infected they have to do double staining for immune cell markers and HIV-1p24.

Authors already have PCR data for the presence of HIV pol sequences in brain tissue. No need to compromise this manuscript.

REVIEWERS' COMMENTS

We would like to thank reviewer 3 for continuing to review this manuscript. Our response to reviewer 3's comments are below in italics.

Reviewer #3 (Remarks to the Author):

The argument “data is data” does not prove that data are correct.

Response: The data are the data and the interpretation is based on what the controls and treatments inform. We can only ask the question, carry out the experiments and then present the data. We interpret these data agnostically and correct or incorrect implies a bias. We simply present the experimental outcome.

The brown depositions detected do not reflect HIV-1p24 presence. Such an artificial signal can happen on paraffin-embedded tissue sections but does not reflect the true presence of protein.

Response: We agree with the reviewer that brown deposits can happen reflecting false positive results. This the reason we have controls which are included in fig 10i,j and fig 11 b to rule out pseudo staining.

Both references #36 and #37 are not correct for positive staining for HIV-1p24 protein.

Reference 36 is a review article and does not provide experimental pieces of evidence of gp120 accumulation in neurons.

Response: reference#36 presents a detailed review highlighting presence of HIV-1 in mice brain, as detected by multiple independent labs. The purpose of this reference is to corroborate our findings with known literature as science is a realized consensus of interpretation and builds on previous studies. It is important to point out to the reviewer that the repeated denial of the utility of hu-CD34+NSG mice models to study brain HIV-1 infection is not warranted when the HIV community has accepted this model, consistently observed HIV p24 in the brain, as we too observed, and in general there have now been dozens of studies carried out with reproducible findings to date using this model.

Reference 37 discuss mechanisms of gp120 internalization in cell culture.

Response: reference#37 shows that even if neurons do not get productively infected by HIV-1, which was not discussed in this article or claimed, that they can still take up gp120 and will stain positive for HIV-1 p24 by IHC. The purpose of this reference is to corroborate our findings with known literature.

I cannot accept provided explanations and would like to recommend removing supplemental figures 10 and 11.

Response: Regrettably we cannot remove supplemental figures 10 and 11 as reviewer#1 and 2 had enquired about proof of presence of HIV-1 in brain as well and both have accepted these data.

If authors claim that immune cells are infected, they have to do double staining for immune cell markers and HIV-1p24.

Response: We do not claim that immune cells are infected with HIV-1 (which is however widely known in literature to be the case, that's why HIV infection causes Immunodeficiencies, because the virus infect immune cells). We only claim that HIV-1 is detectable in brain tissues of HIV-1 infected hu-CD34+NSG mice at 10 weeks post-infection.

Authors already have PCR data for the presence of HIV pol sequences in brain tissue. No need to compromise this manuscript.

Response: It is unclear why presenting the data in a complete and unfettered manner, as we have done, is problematic. We make no claims that are not substantiated by the data presented. Reviewer 3 appears to have serious concerns about the detection of p24 and HIV in the brain. But this is a well-known reported fact of the infection in SCID-CD34 mouse models. WE don't see or report anything different than what others have reported. We would submit that presenting the full data is always preferable as it allows for a thorough unbiased assessment of the scientific work.